# communications
# earth & environment

# Proximity to small-scale inland and coastal fisheries is associated with improved income and food security

Fiona A. Simmance [1,8] ✉, Gianluigi Nico[2,8], Simon Funge-Smith [2], Xavier Basurto [3], Nicole Franz[2], Shwu J. Teoh[1], Kendra A. Byrd [1], Jeppe Kolding [4], Molly Ahern[2], Philippa J. Cohen [1], Bonface Nankwenya[5], Edith Gondwe[6], John Virdin[3], Sloans Chimatiro[5], Joseph Nagoli[5], Emmanuel Kaunda[6], Shakuntala H. Thilsted [1] & David J. Mills[1,7]

Poverty and food insecurity persist in sub-Saharan Africa. We conducted a secondary analysis of nationally representative data from three sub-Saharan Africa countries (Malawi, Tanzania, and Uganda) to investigate how both proximity to and engagement with small-scale fisheries are associated with household poverty and food insecurity. Results from the analysis suggest that households engaged in small-scale fisheries were 9 percentage points less likely to be poor than households engaged only in agriculture. Households living in proximity to small-scale fisheries (average distance 2.7 km) were 12.6 percentage points more likely to achieve adequate food security and were 15 percentage points less likely to be income poor, compared to the most distant households. Households distant from fishing grounds (>5 km) were 1.5 times more likely to consume dried fish compared to households living close. Conserving the flow of benefits from small-scale fisheries is important for meeting the Sustainable Development Goals in the region.

[1] WorldFish, Penang, Malaysia. [2] Fisheries and Aquaculture Division, Food and Agriculture Organization of the United Nations, Rome, Italy. [3] Nicholas School of the Environment, Duke University, Durham, USA. [4] University of Bergen, Bergen, Norway. [5] WorldFish, Lilongwe, Malawi. [6] Lilongwe University of Agriculture and Natural Resources, Lilongwe, Malawi. [7] Australian Research Council Centre of Excellence for Coral Reef Studies, James Cook University, Townsville, Australia. [8] These authors contributed equally: Fiona A. Simmance, Gianluigi Nico. ✉email: Fiona.simmance@gmail.com

Most of the world's extreme poor (433.4 million people) live in sub-Saharan Africa, with 40% of the region's population living below the international income poverty line[1]. One in four people in the region experience undernourishment[2], and 30% of children are stunted due to poor health, inadequate nutrition, and environmental contamination[3]. Poverty and food insecurity have been exacerbated by the Covid-19 pandemic and the impacts of climate change[2]. These drivers impede progress towards Sustainable Development Goals (SDGs) 1 (ending poverty) and 2 (ending hunger); they also affect inter-connected social and environmental development targets[4]. Fisheries, particularly small-scale fisheries, are known to provide important economic and nutrition benefits that can underpin sustainable development and improve access to food for vulnerable populations[5–7]. However, the contribution of fisheries in food systems remains under-valued in policy discourse and development efforts[8,9], globally and in low-middle income countries[10].

Here we analyse the contribution of small-scale fisheries to physical and economic access to nutrient dense food in sub-Saharan Africa. In the region, small-scale fisheries provide the main supply of fish and will continue to do so in the coming decades[11]. Approximately 60 million people depend on coastal and inland small-scale fisheries for their livelihoods[12], which can provide a pathway out of poverty and an engine for rural development[13,14]. Small-scale fisheries also provide an accessible, nutrient dense animal source food for vulnerable populations that increases dietary diversity in the region[15–17]. Substantial knowledge gaps remain, however, in understanding the nature and contributions from small-scale fisheries[18], and who benefits from these contributions[19], particularly at nationally representative scales in sub-Saharan Africa[13].

We present the first novel analyses of a dataset spanning three sub-Saharan Africa nations. We utilised the World Bank nationally representative Living Standard Measurement Survey (LSMS) and its fishery module from Malawi, United Republic of Tanzania (from here on, Tanzania), and Uganda to examine how small-scale fisheries mediate households' physical and economic access to food in urban and rural settings. Fisheries in these countries are primarily associated with the African Great Lakes and smaller inland water bodies, and in the case of Tanzania a 1400 km coastline in the Western Indian Ocean. We matched LSMS georeferenced household-level data with geospatial data on water bodies and coastline location, to examine associations between proximity to water bodies where small-scale fisheries occur (herein referred to as fishing grounds) and small-scale fisheries livelihoods with household poverty, food security, and fish consumption. We discuss our results in the context of the Sustainable Development Goals and regional food systems. The study points to the geographies and parts of society for whom small-scale fisheries must remain a foundation of wellbeing, and we identify areas where novel fish-based interventions are needed to improve food and nutrition security and income gains.

## Results

**Fisheries and poverty**. Across the three countries, 3.4% of households engaged in small-scale fisheries livelihoods across the value chain (harvesting, processing, and trade) as a source of income. Proximity to water bodies increased the likelihood that households were engaged in fisheries. We found (Table 1) that when the distance from fished water bodies increases by 1 km from its mean (33.1 km), the prevalence of income poverty is expected to increase by 0.14 percentage points, given a "baseline" probability of being poor equal to 38.4% in the three countries (baseline probability for poverty 0.3839; $\beta = 0.00136$, $p = <0.01$. Table 1), holding all other variables constant. While this 'per kilometre' distance is small, this translates into an estimated probability of being income poor for households proximate to water bodies that is approximately 15.2 percentage points lower than households living distant from fishing grounds (average distance of 79.3 km) (Supplementary Table 1a), controlling for covariates. While this result presents a correlation rather than indicating causality, it is consistent with other results from the analysis. Results from the same probit regression also suggest that for fishing households, the probability of being income poor was 9 percentage points lower than for farming households, but fishing households were found to have a higher probability (almost 16 percentage points higher) of being poor than households that did not fish or farm (Table 1), all other variables being equal. Positive correlations between small-scale fisheries and income poverty were amplified in contexts where households had poor access to markets and in rural areas. When examining countries individually, this result did not hold for rural Uganda, where fishing households were not better off than agricultural households. In rural Uganda, fishing households were more likely to be income poor by a factor of 9.2 percentage points compared to households that did not fish or farm, and by 3.3 percentage points compared to agriculture households (rural Uganda: for fishing households $\beta = 0.09241$, $p = <0.01$; for agriculture households, $\beta = 0.05965$, $p = <0.01$. Table 1). However, in rural Malawi, fishing households had a lower probability of being income poor not only when compared to agriculture households (−9.3 percentage points) but also when compared to non-agriculture and non-fishing households (Malawi: for fishing household $\beta = −0.08396$, $p = <0.01$; for agriculture households, $\beta = 0.00937$, $p = <0.01$. See Table 1). A higher share of households near fishing grounds and those engaged in small-scale fisheries livelihoods also had higher levels of education (primary level) and asset wealth (total number of assets owned by the household; durables goods and access to infrastructure), but experienced wider dimensions of poverty in terms of marginalisation from access to markets and land (Supplementary Table 6).

**Fisheries and food security**. Overall, the probit regression model revealed that proximity to fishing grounds and engagement in small-scale fisheries was positively associated with adequate food consumption profiles (Food Consumption Scores (FCS)[20]), particularly in rural areas and where households had poor access to markets (Table 2). Fish consumption contributed to the diet of food secure households (3 times more than food insecure households across all countries, T-test, $p < 0.01$), more so than the consumption of food from other animal sources (Supplementary Table 3c). Across all countries, the regression coefficients estimated through the probit model suggest that when the distance from fishing grounds increases by 1 km from its mean (33.1 km), the additive effects on the probability of being food insecure (below adequate food consumption scores) is expected to increase by 0.03 percentage points, given a baseline probability of food insecurity equal to 27% (baseline probability of being food insecure 0.270; $\beta = 0.00032$, $p = <0.01$. Table 2), holding all other variables constant. This implies that the resulting probability of being food insecure (Supplementary Table 1b) for households proximate to water bodies (average distance of 2.7 km) is estimated to be 12.6 percentage points lower compared to households distant from fishing grounds (average distance of 79.3 km) (Supplementary Table 1b). Small-scale fisheries livelihoods were also associated with lower food insecurity, compared to both agriculture households and neither fishing nor agriculture households, by −9.8 and −5 percentage points, respectively and holding all other variables constant (for fishing households $\beta = −0.05019$, $p = <0.01$; for agriculture households $\beta = 0.04790$, $p < 0.01$. Table 2). When examining countries individually, this

**Table 1 Probit regression: estimated marginal effects at the mean distance to water bodies on the probability to be income poor (living below the national poverty line).**

| Variables | All countries | | Malawi | | Tanzania | | Uganda | |
|---|---|---|---|---|---|---|---|---|
| | National | Rural | National | Rural | National | Rural | National | Rural |
| Distance to nearest water body (km) | 0.00136*** (0.00000) | 0.00136*** (0.00001) | 0.00077*** (0.00001) | 0.00061*** (0.00002) | 0.00060*** (0.00001) | 0.00008*** (0.00001) | 0.00164*** (0.00001) | 0.00214*** (0.00001) |
| Distance to nearest water body (km) of households unable to reach food markets | 0.00310*** (0.00002) | 0.00343*** (0.00003) | 0.00931*** (0.00001) | 0.01075*** (0.00002) | −0.00494*** (0.00001) | −0.00419*** (0.00001) | 0.00154*** (0.00002) | 0.00163*** (0.00002) |
| Distance to nearest agricultural market (km) | (0.00002) | (0.00003) | −0.00000 (0.00002) | −0.00294*** (0.00003) | (0.00013) | (0.00013) | −0.00007*** (0.00001) | −0.00075*** (0.00001) |
| Neither fishing nor agriculture HHs, close to water bodies | 0.13008*** (0.00075) | 0.19429*** (0.00116) | 0.14556*** (0.00295) | 0.02641*** (0.00326) | 0.03553*** (0.00093) | 0.02682*** (0.00171) | 0.20714*** (0.00155) | 0.33079*** (0.00239) |
| Fishing HHs, close to water bodies | 0.05244*** (0.00128) | 0.06937*** (0.00149) | 0.02854*** (0.00548) | −0.01141* (0.00609) | 0.00896*** (0.00159) | 0.08416*** (0.00177) | 0.03679*** (0.00246) | 0.04943*** (0.00292) |
| Agriculture HHs, close to water bodies | 0.05363*** (0.00043) | 0.04467*** (0.00050) | −0.07513*** (0.00103) | −0.08115*** (0.00116) | 0.02897*** (0.00062) | 0.02980*** (0.00073) | 0.08171*** (0.00076) | 0.10409*** (0.00092) |
| Households unable to reach food markets | −0.08354*** (0.00098) | −0.07506*** (0.00123) | −0.29404*** (0.00453) | −0.37178*** (0.00552) | 0.45665*** (0.00411) | 0.24855*** (0.00438) | 0.02345*** (0.00086) | 0.04840*** (0.00110) |
| Fishing households | 0.15921*** (0.00086) | 0.06537*** (0.00109) | 0.06462*** (0.00197) | −0.08396*** (0.00251) | 0.21707*** (0.00115) | 0.12305*** (0.00153) | 0.05066*** (0.00187) | 0.09241*** (0.00243) |
| Agriculture households | 0.24929*** (0.00030) | 0.17300*** (0.00058) | 0.18251*** (0.00082) | 0.00937*** (0.00139) | 0.35963*** (0.00042) | 0.27708*** (0.00093) | 0.05557*** (0.00050) | 0.05965*** (0.00074) |
| Household consumed fish (past 7 days) | −0.08574*** (0.00027) | −0.07206*** (0.00032) | −0.24685*** (0.00065) | −0.22931*** (0.00066) | −0.12523*** (0.00040) | −0.07070*** (0.00045) | −0.04466*** (0.00036) | −0.05201*** (0.00046) |
| Household size | 0.03653*** (0.00005) | 0.03951*** (0.00006) | 0.11973*** (0.00019) | 0.13133*** (0.00021) | 0.05730*** (0.00008) | 0.07213*** (0.00010) | 0.01881*** (0.00006) | 0.02654*** (0.00008) |
| Ratio employed household member over not employed | −0.13830*** (0.00044) | −0.14823*** (0.00054) | −0.04342*** (0.00100) | −0.06581*** (0.00108) | −0.21565*** (0.00065) | −0.18829*** (0.00078) | −0.10651*** (0.00070) | −0.08681*** (0.00089) |
| Age of the household head | −0.00008* (0.00004) | 0.00172*** (0.00005) | −0.00723*** (0.00010) | −0.00515*** (0.00011) | −0.00380*** (0.00007) | 0.00275*** (0.00008) | −0.00092*** (0.00007) | −0.00193*** (0.00008) |
| Age of the household head, quadratic | −0.00001*** (0.00000) | −0.00003*** (0.00000) | 0.00006*** (0.00000) | 0.00004*** (0.00000) | 0.00001*** (0.00000) | −0.00006*** (0.00000) | 0.00001*** (0.00000) | 0.00003*** (0.00000) |
| Sex of the head of the household | −0.04366*** (0.00028) | −0.05376*** (0.00034) | −0.05509*** (0.00067) | −0.06445*** (0.00072) | −0.08875*** (0.00041) | −0.09490*** (0.00048) | 0.04974*** (0.00036) | 0.03340*** (0.00049) |
| Education of the head of the household - Primary | −0.07831*** (0.00030) | −0.06867*** (0.00035) | −0.15394*** (0.00072) | −0.14060*** (0.00084) | −0.20624*** (0.00044) | −0.15392*** (0.00048) | −0.01452*** (0.00047) | −0.00136*** (0.00056) |
| Education of the head of the household - Secondary | −0.31264*** (0.00034) | −0.32828*** (0.00048) | −0.32729*** (0.00090) | −0.31063*** (0.00138) | −0.44218*** (0.00052) | −0.35090*** (0.00085) | −0.09635*** (0.00056) | −0.07422*** (0.00075) |
| Wealth index | −0.00035*** (0.00000) | 0.00060*** (0.00000) | −0.07794*** (0.00018) | −0.07426*** (0.00020) | | −0.00507*** (0.00001) | −0.11681*** (0.00022) | −0.14009*** (0.00030) |
| Baseline: average probability to be poor (*100) | 0.384 | 0.478 | 0.446 | 0.517 | 0.443 | 0.593 | 0.212 | 0.239 |
| Distance to nearest water bodies, in km. | 33.1 | 36.0 | 37.0 | 37.1 | 33.8 | 38.0 | 28.5 | 31.2 |
| Observations | 18,610 | 14,275 | 12,444 | 10,174 | 3344 | 1978 | 2817 | 2123 |
| Country/district FE | Yes | Yes | Yes | Yes | Yes | Yes | Yes | Yes |
| r2 | 0.1734 | 0.122 | 0.2539 | 0.2118 | 0.2343 | 0.2077 | 0.1815 | 0.1639 |

Standard errors in parentheses ***$p < 0.01$, **$p < 0.05$, *$p < 0.1$.

**Table 2 Probit regression: estimated marginal effects at the mean distance to water bodies on the probability to be food insecure poor (households with poor food consumption score).**

| Variables | All countries | | Malawi | | Tanzania | | Uganda | |
|---|---|---|---|---|---|---|---|---|
| | National | Rural | National | Rural | National | Rural | National | Rural |
| Distance to nearest water body (km) | 0.00032*** (0.00000) | 0.00074*** (0.00000) | 0.00012*** (0.00001) | 0.00043*** (0.00001) | 0.00046*** (0.00000) | 0.00054*** (0.00001) | 0.00045*** (0.00001) | 0.00061*** (0.00001) |
| Distance to nearest water body (km) of households unable to reach food markets | 0.00229*** (0.00136) | 0.00248*** (0.00182) | 0.00802*** (0.00528) | 0.00737*** (0.00593) | −0.00002 (0.00011) | −0.00034*** (0.00187) | 0.00179*** (0.00388) | 0.00152*** (0.00405) |
| Distance to nearest agricultural market (km) | (0.00002) | (0.00002) | 0.00231*** (0.00018) | −0.00023*** (0.00002) | | | 0.00025*** (0.00001) | −0.00002** (0.00001) |
| Neither fishing nor agriculture HHs, close to water bodies | 0.02227*** (0.00058) | −0.00054 (0.00113) | 0.14017*** (0.00241) | 0.01334*** (0.00287) | −0.00379*** (0.00061) | −0.10137*** (0.00110) | −0.02204*** (0.00072) | 0.02721*** (0.00136) |
| Fishing HHs, close to water bodies | 0.00361*** (0.00136) | 0.09029*** (0.00182) | −0.12426*** (0.00528) | −0.21099*** (0.00593) | −0.09952*** (0.00123) | −0.07870*** (0.00187) | 0.16763*** (0.00388) | 0.19475*** (0.00405) |
| Agriculture HHs, close to water bodies | −0.04130*** (0.00033) | −0.06489*** (0.00040) | −0.01278*** (0.00098) | −0.02707*** (0.00099) | −0.05838*** (0.00041) | −0.09652*** (0.00047) | −0.01754*** (0.00061) | −0.01172*** (0.00065) |
| Households unable to reach food markets | −0.07961*** (0.00075) | −0.09077*** (0.00096) | −0.29026*** (0.00662) | −0.29660*** (0.00770) | −0.09543*** (0.00349) | −0.12966*** (0.00303) | −0.03239*** (0.00062) | −0.02402*** (0.00072) |
| Fishing households | −0.05019*** (0.00107) | −0.09841*** (0.00096) | 0.06895*** (0.00208) | −0.02215*** (0.00234) | −0.05397*** (0.00069) | −0.06445*** (0.00091) | −0.06684*** (0.00232) | −0.07005*** (0.00232) |
| Agriculture households | 0.04790*** (0.00029) | 0.00657*** (0.00054) | 0.18420*** (0.00085) | 0.04203*** (0.00126) | 0.07826*** (0.00034) | 0.07137*** (0.00061) | −0.07963*** (0.00056) | −0.09065*** (0.00080) |
| Household consumed fish (past 7 days) | −0.21989*** (0.00025) | −0.20811*** (0.00030) | −0.26947*** (0.00059) | −0.23625*** (0.00057) | −0.20531*** (0.00031) | −0.17268*** (0.00038) | −0.15220*** (0.00029) | −0.13672*** (0.00033) |
| Household size | −0.01647*** (0.00004) | −0.01794*** (0.00005) | 0.00150*** (0.00017) | 0.00285*** (0.00017) | −0.01905*** (0.00006) | −0.01719*** (0.00007) | −0.00186*** (0.00005) | 0.00009 (0.00006) |
| Ratio employed household member over not employed | −0.04300*** (0.00037) | −0.05893*** (0.00048) | 0.00624*** (0.00097) | −0.01713*** (0.00098) | −0.10211*** (0.00044) | −0.13422*** (0.00059) | 0.10247*** (0.00060) | 0.06795*** (0.00069) |
| Age of the household head | 0.00029*** (0.00004) | 0.00261*** (0.00005) | −0.00727*** (0.00010) | −0.00567*** (0.00011) | 0.00354*** (0.00005) | 0.00980*** (0.00006) | −0.00688*** (0.00006) | −0.00627*** (0.00007) |
| Age of the household head, quadratic | 0.00001*** (0.00000) | −0.00001*** (0.00000) | 0.00008*** (0.00000) | 0.00006*** (0.00000) | −0.00003*** (0.00000) | −0.00008*** (0.00000) | 0.00008*** (0.00000) | 0.00007*** (0.00000) |
| Sex of the head of the household | −0.03137*** (0.00024) | −0.02588*** (0.00031) | −0.01656*** (0.00065) | −0.01281*** (0.00066) | −0.04320*** (0.00030) | −0.02924*** (0.00039) | 0.01471*** (0.00034) | 0.00775*** (0.00040) |
| Education of the head of the household - Primary | −0.07653*** (0.00027) | −0.07210*** (0.00033) | −0.13617*** (0.00072) | −0.11926*** (0.00079) | −0.08226*** (0.00035) | −0.06743*** (0.00041) | 0.00678*** (0.00038) | 0.00634*** (0.00042) |
| Education of the head of the household - Secondary | −0.15468*** (0.00033) | −0.17613*** (0.00045) | −0.30062*** (0.00103) | −0.25736*** (0.00144) | −0.15730*** (0.00041) | −0.13054*** (0.00063) | 0.02853*** (0.00056) | 0.01761*** (0.00068) |
| Wealth index | −0.00196*** (0.00000) | −0.00303*** (0.00001) | −0.05855*** (0.00016) | −0.05456*** (0.00017) | | −0.00345*** (0.00001) | −0.07674*** (0.00019) | −0.10310*** (0.00026) |
| Baseline: average probability to be food insecure | 0.270 | 0.315 | 0.549 | 0.633 | 0.210 | 0.248 | 0.174 | 0.177 |
| Average distance to water bodies, in km. | 33.1 | 36.0 | 37.0 | 37.1 | 33.8 | 38.0 | 28.5 | 31.2 |
| Observations | 18,623 | 14,283 | 12,444 | 10,174 | 3,344 | 1,971 | 2,822 | 2,125 |
| Country/district FE | Yes | Yes | Yes | Yes | Yes | Yes | Yes | Yes |
| r2 | 0.1872 | 0.202 | 0.1829 | 0.1155 | 0.1188 | 0.1449 | 0.1781 | 0.1823 |

Standard errors in parentheses ***$p < 0.01$, **$p < 0.05$, *$p < 0.1$.

result did not hold for Uganda where fishing households were more food insecure than agricultural households but more food insecure compared to neither fishing nor agriculture households (Uganda: for fishing households $\beta = -0.06684$, $p = <0.01$; for agriculture households $\beta = -0.07963$, $p < 0.01$. Table 2). Across the three countries, households engaging in small-scale fisheries livelihoods also experienced lower seasonal food insecurity (a reduction of 0.26 months compared with non-fishing households) (Supplementary Table 6).

Examining dietary patterns using household-level dietary data for a seven-day recall period, we found that fish was the dominant animal source food consumed in urban and rural settings across all countries. A higher percentage of households consumed fish (33–73%, T-test, $p < 0.01$) compared to other animal source foods (<40% eggs and beef, <20% poultry, goat and pork, T-test, $p < 0.01$) (Fig. 1). Malawi and Tanzania had the highest share of households consuming fish (73% and 71% T-test, $p < 0.01$), compared with Uganda where the number of households consuming fish and beef was approximately equal (33% and 32%, respectively, T-test, $p < 0.01$). Fish was proportionately more important in the diets of rural households and the poor, who consumed a lower diversity of animal source foods, less frequently ($p < 0.01$ across all countries. Supplementary Table 3a, b). Fish was mostly acquired through purchases (>95%) especially among households distant to water bodies (97%, T-test, $p < 0.01$, Supplementary Table 4) but with high rates of subsistence fishing (consumption of fish from own production) amongst fishing households close to fishing grounds (41% in Malawi, T-test, $p < 0.01$; 18% in Tanzania, T-test, $p > 0.1$ and 36% in Uganda, T-test, $p > 0.1$) (Supplementary Table 6). Fish, particularly dried fish, was also found to be cheaper and more nutrient dense (e.g., calcium, iron, and omega-3 fatty acids) than other animal source foods, particularly in proximity to small-scale fisheries (Fig. 1 and Supplementary Tables 10 and 12). Although no information is provided on fish species in the surveys, small fish species are known to be dried in the region due to their size which allows efficient drying in time, whilst larger fish species are provided fresh, or either smoked or salted.

Fish consumption patterns also varied by quantity and form (dried or fresh) sub-nationally. Proximity to fishing grounds and engaging in small-scale fisheries were associated with higher quantities (2 times, T-test, $p < 0.01$) and frequencies (2 times T-test, $p < 0.01$) of fish consumption across all countries (Fig. 2 and Supplementary Table 6). Whilst wealthier households typically (highest wealth quintile) consumed twice as much fish as poorer

households (bottom wealth quintile), proximity to fishing grounds lowered the inequalities in the quantities of fish consumed between wealthy and poor households by an average of 30% (Figs. 3 and 4, T-test, $p < 0.01$). However, in Uganda, no association was found which may be because of fishing communities not retaining the benefits of fish (as found in poverty and food security models). Dried fish was the dominant form consumed in Malawi and Uganda (71% of households in Malawi and 64% in Uganda), whilst fresh fish dominated in Tanzania (71%) (Supplementary Table 4). Compared to rural households, a higher share of urban households consumed fresh fish (by 1.3 times), and conversely a higher share of rural households consumed dried fish (by 1.2 times) (Supplementary Table 4). Dried fish was more important to the diets of rural households (by 1.5 times in terms of the share of households, T-test, $p < 0.01$) distant (>5 km) from water bodies compared to those proximate (Fig. 5).

## Discussion

This study is the first to utilize the potential of the LSMS-ISA surveys and its fishery module in applying a food systems lens to understand the value of small-scale fisheries, and the spatial and livelihood determinants of poverty and food security in urban and rural contexts of sub-Saharan Africa. This provides a powerful example of policy-relevant information that can be generated through the efficient approach of augmenting existing survey instruments with fisheries-focussed questions, and in understanding geographical determinants of economic and physical access to food. Overall, the approach provided empirical evidence of the positive association between small-scale fisheries with lower poverty rates, increased fish consumption and improved food security in Malawi, Tanzania and Uganda. A sub-national investigation allowed examination of "where" and "for whom" small-scale fisheries are most important. Small-scale fisheries are the main provider of animal source food and underpin local food systems and sustainable development in the region, shaping food environments and the physical and economic access to food.

Results provide new insights into how proximity to fishing grounds and engagement in fishing activities influence rural economies and contribute to SDG 1 – no poverty. Households living close to fishing grounds were able to spend more money to meet their basic needs. Fishing grounds were largely located in rural areas, with the flow of economic benefits from small-scale fisheries amplified in these contexts. The association between fishing grounds and prevalence of poverty was however small, and our study did not investigate the many other socio-economic drivers of poverty and geo-location determinants (e.g., agency, mobility, land access, socio-culture, infrastructure) that influence wealth generation near fishing grounds and how those living far away exit poverty. Small-scale fisheries livelihoods were associated with lower poverty rates in Malawi and Tanzania, in accordance with other studies[21–23], however this relationship was not found in Uganda. This may reflect a difference in how fishing households were defined in the LSMS-ISA surveys. In Uganda only households engaging in harvesting of fish were defined as fishing households, while in Malawi and Tanzania the definition included processing and trading, activities which usually generate higher income[13,22]. Uganda's fisheries are also mainly from Lake Victoria[24], where the Nile Perch commercial fishery dominates economically with export value-chains that drive inequity in benefits retained for local fishing communities[25,26]. Households engaged in small-scale fisheries face wider dimensions of poverty beyond income. Consistent with other studies, we show that land access and asset wealth varies by contexts[27,28], and small-scale fisheries are often marginalised from economic services such as access to agricultural markets[29].

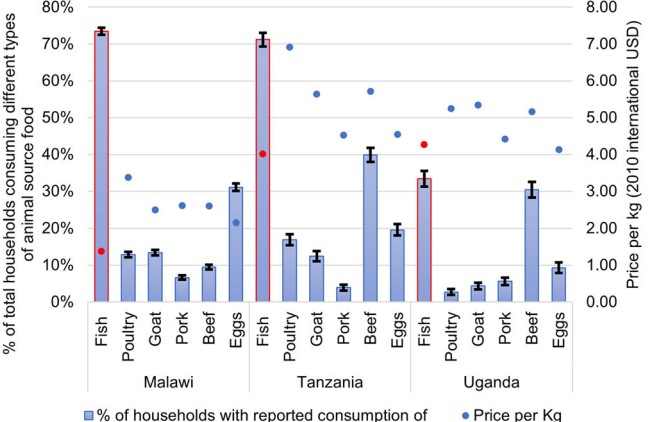

**Fig. 1 Share of households (% of total) consuming animal source foods and prices of purchased food (average price per kilogram in international US$).** Columns represent food consumption and dots depict prices, with red highlighting fish. Lines show 95% confidence interval (mean ± standard error).

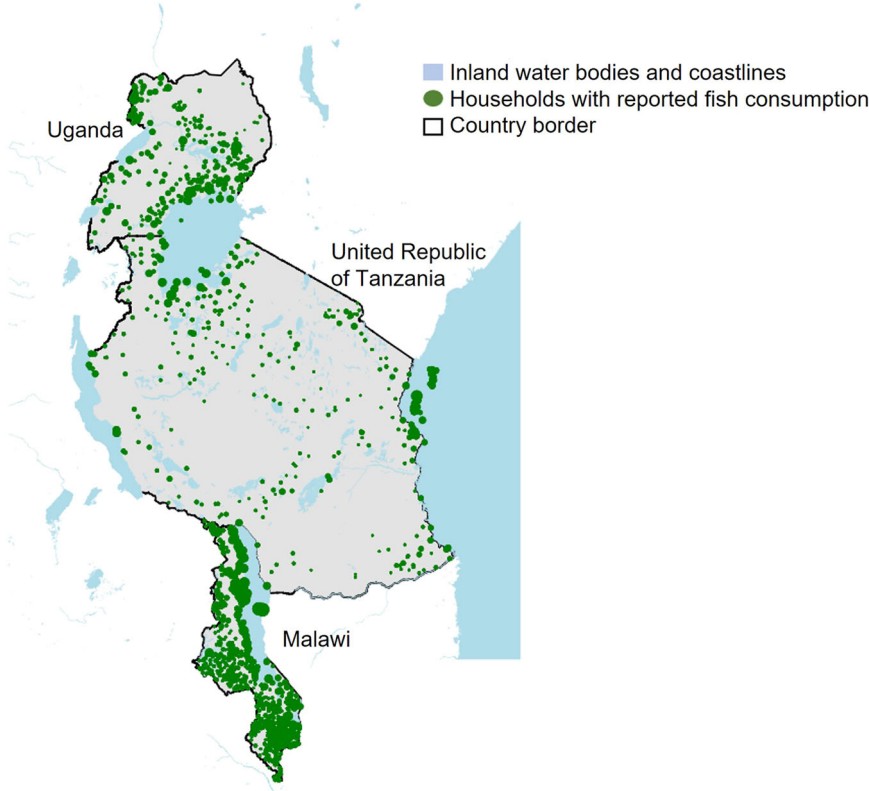

**Fig. 2 Spatial distribution of households reporting any fish consumption (green dots) by open inland water bodies (≥0.1 km²) and coastlines for Uganda, Tanzania and Malawi.** Analysis is of 18,715 households; a sample framed to be representative of the total population of each country (93.8 million population in total for all three countries). Data source: the World Bank's Living Standards Measurement Surveys and Integrated Surveys on Agriculture (LSMS-ISA) for Malawi (2016–17), Tanzania (2014–15) and Uganda (2010–11); Global Lakes and Wetlands Database (GLWD)[45], and the European Space Agency GlobCover databases for coastlines[46].

Fish was found to be not just the most frequently consumed animal source food across all countries, but also the most nutrient dense and affordable; particularly dried small fish eaten whole, supporting other studies of diets in the Africa Great Lakes region[30,31]. As expected, proximity to fishing grounds increased fish consumption, but importantly lowered inequalities in consumption between poor and wealthy households, particularly for the rural; contributing to SDG 10 – reduced inequalities. Studies have also shown fish consumption to be higher in fishing communities in the region, such as in Lake Ruwe in Tanzania[16], Lake Chilwa in Malawi[22] and in coastal fisheries in Kenya[15]. The importance of trade in distributing the benefits of small-scale fisheries was illuminated through new data on consumption patterns, showing dried fish to be important in rural and urban areas, but particularly so in areas distant from fishing grounds. Limited empirical evidence exists on the value of dried small fish in the region; such as in urban[32] and rural environments[22], and our study provides novel insights at broader scales on its role in improving access to nutritious food for vulnerable populations[30,33]. Small fish, primarily from inland small-scale fisheries, contribute around 70% of total catches in the regions, and this proportion is steadily increasing[34].

Our findings also revealed that small-scale fisheries play a critical role in improving physical and economic access to food in contexts where traditional food systems dominate, and formal markets fail to penetrate. Engagement with small-scale fisheries livelihoods and proximity to fishing grounds had a positive association with food security and the household food consumption scores; supporting SDG 2 – ending hunger. Most fish was purchased illuminating the importance of fish for economic

access to food, fish as an income pathway and the extent of trade. In addition, proximity to fishing grounds influenced physical access to fish as food, subsistence fishing, and prices of fish at markets. Two other studies in the region support our finding that access to markets is not the main driver of diversity of food consumption, rather access to resources including small-scale fisheries, is more important[17,22]. However, in rural Uganda, no association between small-scale fisheries and food security was found in our study. Again, this could be a result of the inequity in the flow of benefits from export-orientated value-chains in Uganda[25,26].

Our findings highlight where, and for whom, hunger, poverty and low rates of fish consumption prevail, and provides empirical evidence of the diversity of small-scale fisheries and their importance in food systems[35]. There is clearly an access issue for a high proportion of the rural poor distant from fishing grounds, and particularly for those who don't fish. In Uganda, 79% of poor households and 71% of rural households living distant from fishing grounds did not consume fish, and inequalities in fish consumption were higher for households close to fishing grounds. This demonstrates the limits of the penetration of fish trade and the trade-offs with an export-oriented fishery in Uganda[25,26]. These numbers were lower in Malawi (33% rural and 40% poor) and Tanzania (34% rural and 38% poor), with well-developed domestic and regional fish trade routes[34]. Rural and poor households distant from fishing grounds consumed some of the lowest quantities of fish. However, dried fish were found to be most accessible to these remote communities and were often the main accessible animal source food. Strategies are needed to leverage the benefits of small-scale fisheries across value-chains,

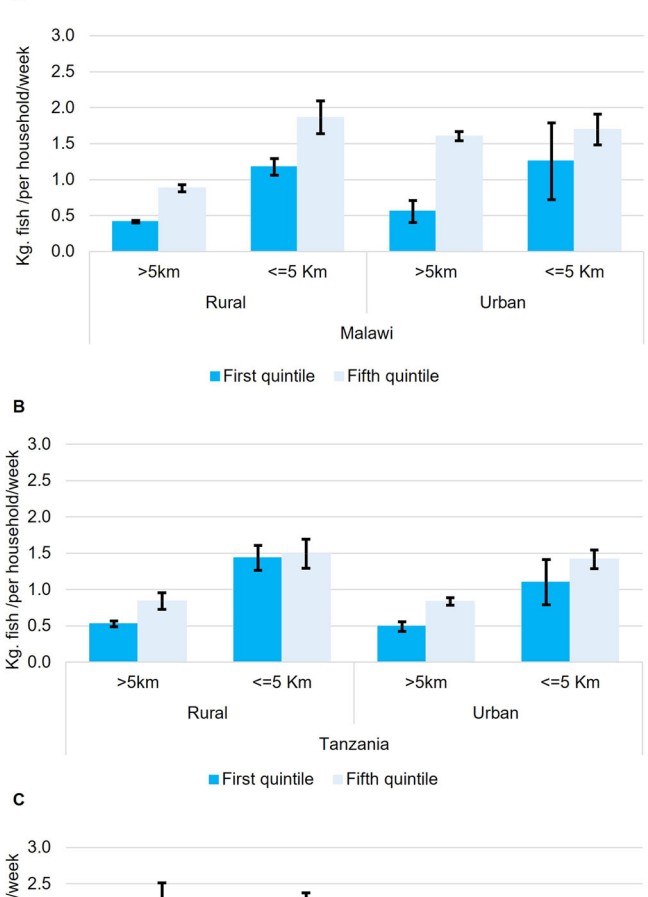

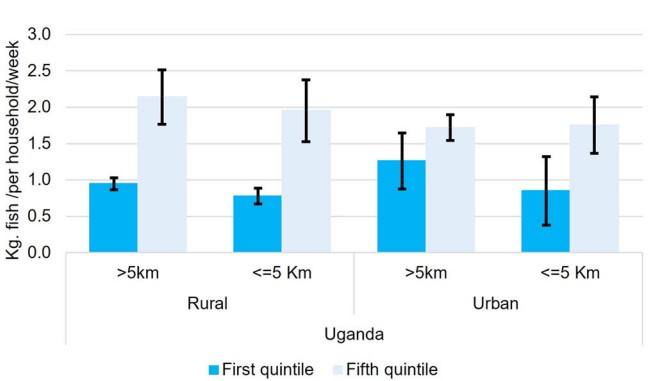

**Fig. 3 Fish consumption (kg/household/week) by richest (fifth quintile) and poorest (first quintile) households.** Graph depicts consumption patterns in rural and urban areas that are proximate (<5 km) or distant (>5 km) from water bodies in (**A**) Malawi, (**B**) Tanzania and (**C**) Uganda. Lines show 95% confidence interval (mean ± standard error).

such as prioritising local trade policies to ensure equitable flow of benefits to fishing communities, and distribution of dried small fish to remote vulnerable groups. This corroborates findings by O'Meara et al.[17] who showed that children living distant from fishing grounds in Malawi and Zambia have the lowest access to fish as food, and that access as well as equitable intra-household distribution of food is needed to tackle malnutrition.

Our findings shed light on the important role of small-scale fisheries for sustainable development in the region, and notably for leaving no one behind in meeting the Sustainable Development Goals relating to 1 (ending poverty), 2 (ending hunger), 10 (reducing inequalities) and 14b (access for small-scale fishers to marine (and inland) resources and markets), and securing the rights of fishers under the Voluntary Guidelines for Securing Sustainable Small-Scale Fisheries. This means that small-

fisheries resources remain critical to improving food and nutrition security outcomes and policies and interventions must look beyond the production sector, engaging locally in food systems to identify pathways across food environments that enhance the flow of benefits to consumers in rural and urban settings. This situation is by no means static and as populations, economies, and infrastructure change, the productivity and viability of these small-scale fisheries will also be affected. Fishing communities, although not always income poor, can be increasingly vulnerable to multiple shocks, such as habitat degradation[36], outdated regulations[37] or climate variability that erodes their asset wealth[38]. Across Malawi, Tanzania and Uganda, a 1% increase in fish supply, such as via reducing waste and loss, or better utilisation of the lightly exploited but highly productive small species[34], could enable almost 250,000 additional people to meet the minimum recommended intake of 28 g of fish per day for a healthy and sustainable diet (Supplementary Table 9). Improvements in fisheries governance (to prioritise benefits to local communities), post-processing practices and the trade and distribution of fish could enable more fish to reach vulnerable rural populations distant from fishing grounds, and can increase the resilience of fisheries. Furthermore, the LSMS national survey provides a unique and valuable data source for illuminating the contribution of small-scale fisheries to sustainable development[9,39]. Nations globally could adopt similar survey approaches to value aquatic foods in food systems, with further research opportunities. In Africa's Great Lakes and coastal Western Indian Ocean region, policies that proactively recognise and integrate the value of fish and small-scale fisheries for sustainable development will safeguard these important benefits amidst increasing pressures and drivers. In order to enhance the flow of benefits from small-scale fisheries in vulnerable rural contexts, more research is needed to examine the distributional benefits between fishing grounds, where best management and target approaches can be identified.

## Conclusions

Income poverty and food insecurity remain unacceptably high in sub-Saharan Africa. With systematic underreporting and historic neglect and denigration, the contribution and relationship of small-scale fisheries to sustainable development is poorly understood. Our findings highlight that through provision of fish as food and income, small-scale fisheries improve physical and economic access to food for both urban and rural populations in Malawi, Tanzania and Uganda. We highlight evidence that proximity to fishing grounds, and engagement in small-scale fisheries livelihood activities, are associated with lower income poverty and higher food consumption profiles across countries. The findings illuminate the often-obscured value of inland and coastal small-scale fisheries in the African Great Lakes region and Western Indian Ocean for nutrition and livelihoods. Importantly, small-scale fisheries provide an important role for helping countries achieve the Sustainable Development Goals, and their neglect as a foundational support to nutrition security would not only undermine progress towards sustainable food systems, but also constrain forward momentum in improving food and nutrition security of the most vulnerable. We therefore call for greater recognition of the value of small-scale fisheries in policies, and more sustainable and inclusive governance and management of the sector to promote the flows of benefits to vulnerable populations.

## Methods

**Study design.** We used a food systems framing to conceptually position our research to investigate how small-scale fisheries shape two key aspects of food environments - physical access to food via living in proximity to small-scale fisheries (fish as food pathway), and economic access to food via small-scale fisheries livelihoods (fish as income pathway).

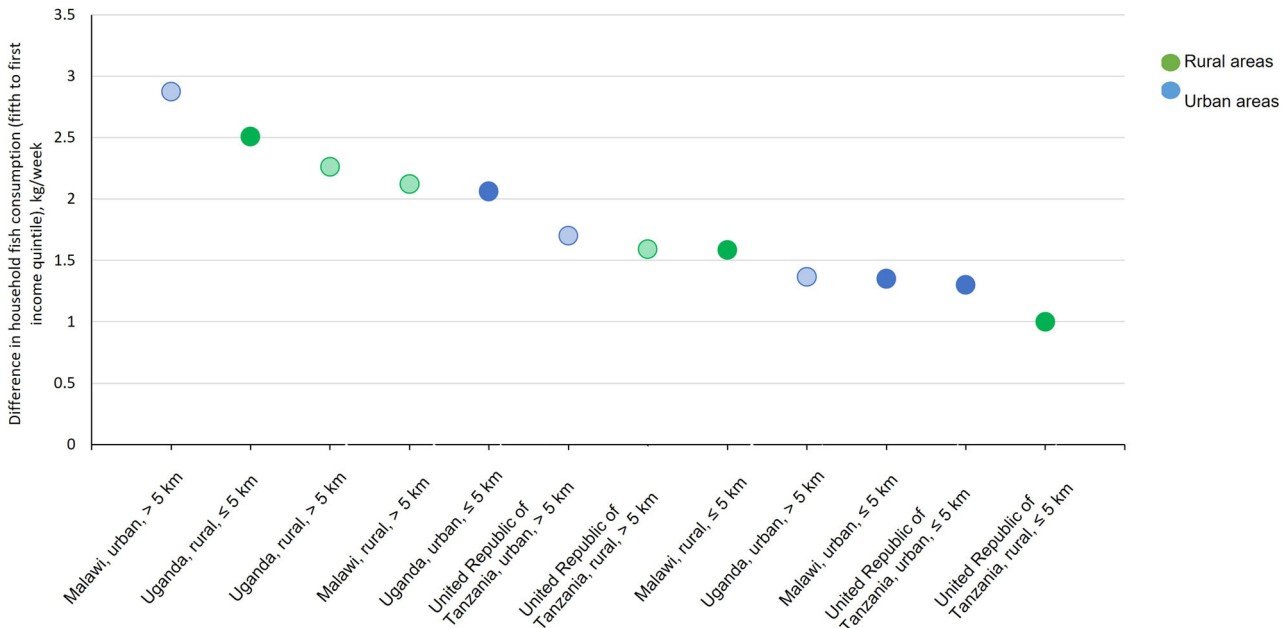

**Fig. 4 Inequalities in fish consumption sub-nationally.** Ratio in the quantities of fish consumed (kg/household/week) between richest (fifth quintile) and poorest (first quintile) households, by rural and urban areas and proximate (<5 km) or distant (>5 km) from water bodies in Malawi, Tanzania and Uganda. Graph depicts greater inequalities in consumption of fish in contexts on the left, and lower inequalities on the right. Solid colours represent proximate to fishing grounds and faded distant.

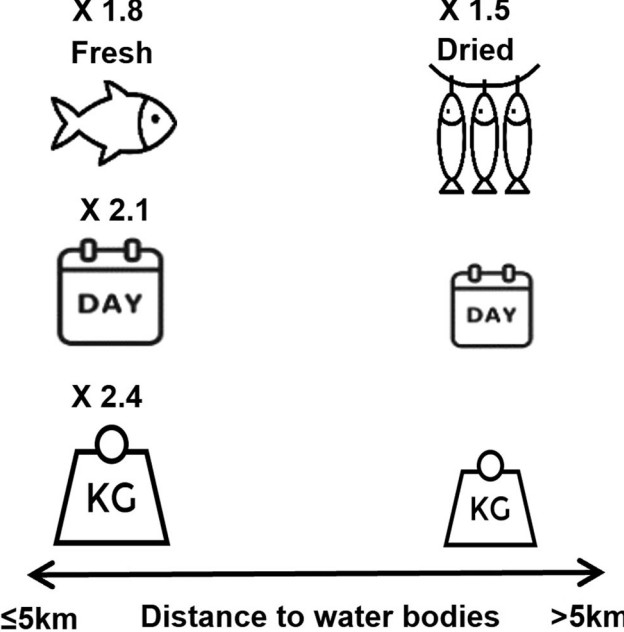

**Fig. 5 Fish consumption by proximity to water bodies across rural Malawi, Tanzania and Uganda.** Representing the average difference in the share of total households consuming dried and fresh fish, quantity of fish consumed (kg/household/week) and frequency (number of days out of 7 days).

We examined food system components of supply chains (small-scale fisheries livelihoods related to harvesting, processing and trade), food environments (proximity to small-scale fisheries and livelihoods), income poverty status, and household diets (fish consumption and annual food security) (Supplementary Fig 7)[40,41]. Small-scale fisheries are notably recognised for their safety net function during times of shocks and extreme events, increasing the ability of households to recover, exit poverty and afford food over the longer-term[42].

**Country selection and household survey data.** We selected Malawi, Tanzania and Uganda, given these countries represent a region where small-scale fisheries provides the main supply of fish and are important for rural inland and coastal livelihoods[24,43], and yet substantial data gaps remain in valuing small-scale fisheries in the regional food system. Small-scale fisheries, particularly inland fisheries, in this region are known to be highly productive with a linear increasing trend in catches over the last three decades[25,35]. On average 70% of the total catches consist of small pelagic species, which are largely driven by climate, and are highly productive, resilient, and under-exploited[34]. However, challenges do exist in fisheries governance and signs of over-exploitation of some few fish stocks[44], as well as high post-harvest fish waste and loss across value-chains undermine the potential benefits from the sector[23]. We analysed the World Bank's Living Standards Measurement Surveys and its Integrated Surveys on Agriculture (LSMS-ISA) from Malawi, Tanzania and Uganda. The LSMS-ISA surveys conducted in these countries collected georeferenced household-level data and had been designed and implemented with a dedicated fishery module[39] which contained questions on household fish consumption (frequency, quantity, and form of fresh or dried fish) and small-scale fisheries livelihoods across value chains (harvesting, processing and trading). The fishery module was collected across different years in Malawi (2016–17), Tanzania (2014–15) and Uganda (2010–11), and accordingly these are the years analysed in this study. The LSMS-ISA surveys collects consumption data over a period of 12 months so that the indicator captures the intrinsic variability due to seasonality, such as low and high periods of food consumption.

**Geospatial data and distance to fishing grounds.** Georeferenced household data from LSMS-ISA surveys were matched with geospatial data on the location of inland water bodies and coastlines (Supplementary Table 11) to investigate geographic correlates (e.g., distance to fishing grounds - water bodies where fisheries occur) of poverty and food security. Data on inland water bodies were from the Global Lakes and Wetlands Database (GLWD)[45], and the European Space Agency GlobCover databases for coastlines[46]. Inland water bodies from the GLWD database include permanent, open water bodies (e.g., lakes, reservoirs, rivers) with a surface area ≥0.1 km$^2$ for each country, including cross-border water bodies. We selected water bodies to represent types of water bodies known to support fisheries, based on catch data[24,43]. We assume the entire coastline of Tanzania was accessible and used for marine small-scale fisheries. We use the term 'water body' to mean either freshwater or marine waters.

Distance between water bodies and households was calculated as the shortest, straight line, distance from the household location (identified through the GPS coordinates of the households) to any point of the nearest water body. The distance was expressed in km.

In our descriptive statistics, a cut-off threshold of 5 km from fishing grounds was used to compare the key indicators presented in this study (e.g., percent of poor and food insecure households, frequency and quantity of fish consumption, etc), for households proximate and distant (≤5 km was considered close and >5 km

was considered far) from fished water bodies, as well as between fishing and non-fishing households. The choice of the cut-off threshold used for our descriptive statistics was guided by other studies[16,17], in addition to reflecting the distribution of households by quintile of distance to water bodies. Concerning the latter, we found that the average distance from fishing ground of the first quintile was always lower than 5 km in all countries.

In the regression analyses, the distance to water bodies was included as a continuous variable (in km). This choice reflects the need to better understand dynamics for households that tend to live more distant from fishing grounds. These dynamics were captured by measuring the marginal increase in the probability of being poor or food insecure for a one-unit increase (1 km) from the mean distance to fishing grounds.

We acknowledge two limitations behind the calculation of the straight-line distance to water bodies. First, using the straight-line distance to water bodies may introduce biases in the statistical analyses presented, especially for households located in any particular landscapes within the country. The walking or travel time distance over a road network would provide a better alternative, however there is lack of data on road networks. Despite the straight-line distance to water bodies encompasses some limitations, we still believe that this method of calculation provides a good proxy to categorize household in relation to their distance to water bodies, and the results from the analyses should not deviate substantially from other method of calculation. For example, a study[51] found that the straight-line distance tends to be highly correlated (R > 0.91) with both walking and travel time distance.

Second, an additional bias in the presented analyses may be introduced due to the modification strategy of the households GPS coordinates. This strategy was implemented before dissemination of household level data to avoid the risk of disclosure of sampled households. In its essence, the modification strategy relies on random offset of cluster center-point within a specified range. For urban areas a range of 0–2 km is used. In rural areas, where risk of disclosure may be higher due to more dispersed communities, a range of 0–5 km offset was used. While we had no control over this modification strategy, we believe that the modification of the GPS coordinates does not affect the way households are classified in relation to their distance to fishing grounds: considering that the modification strategy was applied to both distant and proximate households, we expect that the distribution between households close and distant to water bodies has remained unchanged and, hence, the presented statistics are still valid for the analysis.

**Variable construction.** We used a range of socio-economic indicators across food system components (Supplementary Table 11). As a measure of physical and economic access to food we used two indicators of small-scale fisheries: proximity to fishing grounds and fishing households. Household livelihoods were assigned according to whether households primarily, but not exclusively, engaged in small-scale fisheries (fishing, harvesting, processing and/or trading which varied by survey), agriculture (e.g., crop or livestock), or neither fisheries or agriculture. For each country survey, households were categorised according to their engagement in fishing and/or agriculture activities in the prior 12 months. Households in which one or more member engaged in fish-related activities were defined as 'fishing households'. Fish-related livelihood activities were defined as fish harvesting, processing, and trading in Malawi and Tanzania, whilst in Uganda they were defined only as fishing. Households with one or more member engaged in agriculture, but not in fish-related activities, were defined as 'agriculture households'. Through data exploration of livelihood categories, we found that 96% of all fishing households in our study combine fish-related and agricultural activities, with only 4% engaging exclusively in small-scale fisheries. Examination of diverse livelihood typologies within fishing household category (e.g., fisher-farmer, which is common in the region or exclusive fisher) was deemed out of the scope of this study and not feasible due to the small number of observations of exclusive fishers.

Household poverty was measured using the per-capita monthly expenditure (equivalized using the adult equivalent scale). Poor households were defined as those households with a per-capita monthly expenditure below the national poverty line. The national poverty line –which was defined by national authorities in the three countries analysed– is a country-specific monetary threshold below which a household (and its members) cannot meet their basic needs. The poverty metric, as defined above, was used across physical, natural and human capital: asset wealth, distance to markets, access to land and education level of head of household. Since the asset wealth captures the typologies and number of assets owned by the household (durable goods - radio, bicycle, TV; utilities and infrastructure – access to protected water source and electricity), we developed an index for assets using the principal component analysis. This technique reduced the multi-dimensionality of the asset's variables, and it allowed the data to identify the linear combinations of the assets components that explain the greatest share of the variation in wealth. As the final wealth index was standardised across households, this index allowed providing a ranking of households which reflected their ownership of assets.

Food security was measured using two indicators; household-level food consumption profile – using the Food Consumption Score (FCS) index[20], and subjective food insecurity defined as the number of months during a year that a household reported not having enough food to feed the household. Together, these indicators provide a more comprehensive understanding of household food

security over a longer period than other surveys (e.g. Demographic and Health)[47–49]. The LSMS-ISA surveys collects food consumption data over a 7-day recall period. To capture seasonal variation in the food consumption indicators, sampled households were interviewed over a 12-month period: for each month of the year, a different portion of sampled households was interviewed so that the derived indicators reflect the intrinsic variability in food consumption, which may be due to seasonality. We used the FCS index as a food security indicator as it is akin to the data collected via the LSMS-ISA surveys, and that there was a need for comparison across select countries. The FCS index measures the frequency (number of days) and diversity of food groups consumed over a 7-day recall period, with weights given to groups based on nutritional value. The FCS is validated as a proxy for energy sufficiency (quantity of food) and food access, and is associated with other household-level diet diversity measures (e.g. household dietary diversity score (HDDS))[20,48]. The difference between FCS and other indicators such as HDDS is the recall period (7-days versus 24 h), diversity of groups, weights assigned based on nutrition, and use of frequency together with diversity of groups consumed. The FCS with a longer recall period can show more habitual consumption but can also have limitations with people's recall reliability. Although it has not been validated yet as an indicator for micronutrient intake, it does provide weights to nutrient-rich food groups and accounts for frequency of consumption, which other indicators do not. Fish consumption was described in terms of the (i) quantity (kg of wet weight equivalent per household per week), (ii) form (fresh, dried, smoked, other) and (iii) source (purchased, own consumption, gift) of fish consumed. The share of households reporting consumption of other animal source foods was also calculated to examine the relative role of fish in overall diets.

We also examined the prices of foods consumed to investigate the accessibility of fish as food in terms of affordability compared to animal source foods. The LSMS-ISA survey collects data on the value and volume of food that were purchased and consumed. Those two variables were further used to construct the average price for each food item. To control for price level differences between countries, food prices data calculated from the survey were converted from local currency unit to international USD, using the Purchase power parity conversion factor corresponding to the year of the survey (Source: World Development Indicators database, World Bank). Moreover, since the surveys were conducted in different years, nominal prices corresponding to the years of the surveys were converted into real, inflation-adjusted prices using the Consumer Price Index (CPI, base year: 2010). This allowed to control for potential inflation patterns within countries and provide a better comparison of food prices per Kg. across the three countries analyzed (Source: World Development Indicators database, World Bank).

Finally, we drew upon nutritional databases (food composition tables, FishBase and Illuminating Hidden Initiative) to understand the relative nutritional value of fish; by species, size (small or large) and form (e.g., fresh or dried), compared to other animal source foods (Supplementary Table 12). This enables us to contextualise the nutritional importance of consumption patterns.

**Descriptive statistics.** We created a harmonized multi-country dataset for Malawi, Tanzania and Uganda with 18,715 nationally representative household-level observations. The sample included in this study represents more than 19 million households corresponding to a population of 93.8 million people across the three countries (Supplementary Information).

Descriptive statistics were calculated to compare poverty and food security indicators among households proximate and distant from fished water bodies, and between fishing and non-fishing households (see full details in Supplementary Information). For this analysis, households distant and proximate from fished water bodies were clustered into two groups based on a cut-off threshold of 5 km (distant > 5 km; proximate ≤5 km). The Welch's t-test was then used throughout to assess the statistical significance of mean statistics between these two groups.

**Econometric model.** The estimated probabilities of being poor (household living below the national poverty line) and food insecure (household with a poor food consumption profile) were modelled through two separate probit regression models, where the outcome variable was equal to 1 for poor and food insecure households and 0 otherwise. The independent variables in both models included the household's distance to water bodies and the distance to food market. Both variables are expressed as continuous variables (in km), reflecting the need to measure the marginal increase in the probability of being poor or food insecure (i.e., the estimated β coefficients) given a one-unit change (1 km) in the distance to fishing ground (or food markets) from its mean. Both models also included an interaction variable which measured the household's distance to water bodies but restricted to only those households who were unable to reach the food market. We tested this interaction as we expected that living in proximity to water bodies could mitigate the negative effects on poverty and food insecurity when households are unable to access food markets. In order to measure the conditional difference in the average probability to be poor and food insecure between households who engaged in fisheries and households who engaged in other non-fishing activities, we constructed a categorical variable that classified households according to their main livelihood activity, namely (1) neither fishing, nor agriculture households (i.e., the reference baseline household category), (2) fishing households and (3) agriculture households. This categorical variable was further restricted to only households

living in proximity to water bodies to better measure for which typology of household the proximity to fishing grounds is most beneficial. Both models were controlled for the age, sex and the highest level of education attained by the head of the household, as well as the total number of household members employed (over total household members) and the wealth index of the household.

For each model (poverty and food insecurity), we examined associations at the cross-country, national and rural levels (Tables 1 and 2, also available as Supplementary Data 1 and 2). Stata 15 was used for all statistical analyses. Both descriptive statistics and the regression coefficients were estimated using the household probability weight, the latter instrumental to make the derived indicators from the surveys representative of the population of interest thus allowing general inference for the three countries.

**Reporting summary**. Further information on research design is available in the Nature Research Reporting Summary linked to this article.

## Data availability

The full data tables from our econometric model are publicly available on Figshare: Table 1: https://figshare.com/articles/dataset/Supplementary_Data_Table_1a_docx/ 19898002 Table 2: https://figshare.com/articles/dataset/Supplementary_Data_Table_1b_ docx/19898005. Data analysed for this study was sourced from the following: the World Bank's Living Standards Measurement Surveys and Integrated Surveys on Agriculture (LSMS-ISA) for Malawi (2016–17), Tanzania (2014–15) and Uganda (2010–11).

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

## Acknowledgements

This work was undertaken within the context of the Illuminating Hidden Harvests (IHH) study conducted by the Food and Agriculture Organization (FAO), in partnership with WorldFish and Duke University. Additional support was provided by the CGIAR Research Program (CRP) on Fish Agri-food Systems (FISH), led by WorldFish, supported by contributors to the CGIAR Trust Fund. The designations employed and the presentation of material in this information product do not imply the expression of any opinion whatsoever on the part of the Food and Agriculture Organization of the United Nations (FAO) concerning the legal or development status of any country, territory, city or area or of its authorities, or concerning the delimitation of its frontiers or boundaries. The views expressed in this information product are those of the authors and do not necessarily reflect the views or policies of FAO. This paper has not gone through the standard science-review procedure of the FAO.

## Author contributions

F.A.S. co-led the study and conceived the idea, developed the design and wrote the article. G.N. co-led the study and conceived the idea, contributted to the design and conducted the analyses. S.F.S. conceived the idea, contributed to the design, conclusions and approved the final manuscript. X.B. conceived the idea, contributed to the design, conclusions and approved the final manuscript. N.F. conceived the idea and contributed to the conclusions and approved the final manuscript. S.J.T. contributed to the GIS analyses, methods and approved the final manuscript. K.A.B. contributed to the methods, conclusions and approved the final manuscript. J.K. contributed to the conclusions and approved the final manuscript. M.A. contributed to the conclusions and approved the final manuscript. P.J.C. contributed to the conclusions and approved the final manuscript. B.N. contributed to the conclusions and approved the final manuscript. E.G. contributed to the conclusions and approved the final manuscript. J.V. contributed to the conclusions and approved the final manuscript. S.C. contributed to the conclusions and approved the final manuscript. J.N. contributed to the conclusions and approved the final manuscript. E.K. contributed to the conclusions and approved the final manuscript. S.H. Thilsted contributed to the conclusions and approved the final manuscript. D.J.M. contributed to the design, conclusions and approved the final manuscript.

## Competing interests

The authors declare no competing interests.
