## [Peer Review File · Communications Earth & Environment]

Web links to the author's journal account have been redacted from the decision letters as indicated to maintain confidentiality.

2nd Mar 22

Dear Dr Simmance,

Your manuscript titled "Proximity to small-scale inland and coastal fisheries is associated with improved income and food security" has now been seen by 3 reviewers, and I include their comments at the end of this message. They find your work of interest, but some important points are raised. We are interested in the possibility of publishing your study in Communications Earth & Environment, but would like to consider your responses to these concerns and assess a revised manuscript before we make a final decision on publication.

We therefore invite you to revise and resubmit your manuscript, along with a point-by-point response that takes into account the points raised. Please highlight all changes in the manuscript text file.

In addition to the reviewers' comments, please ensure that the revised manuscript addresses the following points:

1. Provide a compelling assessment of the contribution of small-scale fisheries to food security in the Great Lakes region including an investigation on the influence of seasonality.
2. Provide an assessment, or at least a thorough discussion, of the ecological sustainability and potential stresses on the fisheries resources in the Great Lakes region
3. Ensure that your statistical analysis is robust and discuss the statistical significance of the results obtained

Please use the following link to submit your revised manuscript, point-by-point response to the referees' comments (which should be in a separate document to any cover letter) and the completed checklist:

[link redacted]

We hope to receive your revised paper within six weeks; please let us know if you aren't able to submit it within this time so that we can discuss how best to proceed. If we don't hear from you, and the revision process takes significantly longer, we may close your file. In this event, we will still be happy to reconsider your paper at a later date, as long as nothing similar has been accepted for publication at Communications Earth & Environment or published elsewhere in the meantime.

We understand that due to the current global situation, the time required for revision may be longer than usual. We would appreciate it if you could keep us informed about an estimated timescale for

resubmission, to facilitate our planning. Of course, if you are unable to estimate, we are happy to accommodate necessary extensions nevertheless.

Please do not hesitate to contact me if you have any questions or would like to discuss these revisions further. We look forward to seeing the revised manuscript and thank you for the opportunity to review your work.

Best regards,

Gerald Forkuor, PhD
Editorial Board Member
Communications Earth & Environment

Joe Aslin
Senior Editor
Communications Earth & Environment

EDITORIAL POLICIES AND FORMATTING

Editorial Policy: [Policy requirements](https://www.nature.com/documents/nr-editorial-policy-checklist.zip)

Furthermore, please align your manuscript with our format requirements, which are summarized on the following checklist:

[Communications Earth & Environment formatting checklist](https://www.nature.com/documents/commsj-phys-style-formatting-checklist-article.pdf)

and also in our style and formatting guide [Communications Earth & Environment formatting guide](https://www.nature.com/documents/commsj-phys-style-formatting-guide-accept.pdf) .

*** DATA: Communications Earth & Environment endorses the principles of the Enabling FAIR data project (<http://www.copdess.org/enabling-fair-data-project/>). We ask authors to make the data that support their conclusions available in permanent, publically accessible data repositories. (Please contact the editor if you are unable to make your data available).

All Communications Earth & Environment manuscripts must include a section titled "Data Availability" at the end of the Methods section or main text (if no Methods). More information on this policy, is available at <http://www.nature.com/authors/policies/data/data-availability-statements-data-citations.pdf>.

DATA SOURCES: All new data associated with the paper should be placed in a persistent repository where they can be freely and enduringly accessed. We recommend submitting the data to discipline-specific, community-recognized repositories, where possible and a list of recommended repositories is provided at http://www.nature.com/sdata/policies/repositories.

If a community resource is unavailable, data can be submitted to generalist repositories such as figshare or Dryad Digital Repository. Please provide a unique identifier for the data (for example a DOI or a permanent URL) in the data availability statement, if possible. If the repository does not provide identifiers, we encourage authors to supply the search terms that will return the data. For data that have been obtained from publically available sources, please provide a URL and the specific data product name in the data availability statement. Data with a DOI should be further cited in the methods reference section.

Please refer to our data policies at http://www.nature.com/authors/policies/availability.html.

REVIEWER COMMENTS:

Reviewer #1 (Remarks to the Author):

The paper wishes to present relationship between fishery/poverty and food. Despite using a large database, the results are not significant from a statistical and empirical point of view. For instance the 1.2% more poverty every 10km distance from the shores is a non sens and doesn't mean something at the household level. Same with the other figures (food insecurity ... 0.3 and 0.6%). Peasants are entering the fishery when drought or other events are altering crops and most of them are seasonal fishermen (at the end of the raining season) but they do come back to their land. So, overall, due to the lack of a strong ground anchorage, the paper provides generalities that are already known and provdes little for policy implication.

Reviewer #2 (Remarks to the Author):

Dear authors,

Thanks for the oppportunity to review your interesting manuscript. By drawing on food systems concepts and frameworks, and harnessing large nationally-representative datasets, this manuscript responds to key knowledge gaps about patterns of fish intake and food security in a region with potential for investment and innovation in small-scale fisheries to contribute to combatting the

ongoing high prevalence of undernutrition. The manuscript is well-written, presents novel analyses, contextualises these within the wider literature, and proposes their significance for research and development efforts.

Please consider these specific methodological points, which I believe would increase rigour, improve clarity and acknowledge limitations:

Categorising livelihoods

- 290. Why have mixed livelihood strategies been excluded from your analysis? It is acknowledged that households commonly combine fishing and agricultural activities, which makes sense given the seasonal variation in income and labour for both. It seems unwise to exclude these households from your sample, given the value of understanding whether mixed livelihood strategies offer greater resilience to seasonal food insecurity. Why not consider these as an additional livelihood group / typology within your analyses?
- Also worth adding a sentence or two to better define livelihood categories in the methods section. i.e. at least one household member involved in fishing / agriculture for any period of time within the past 12 months? In the discussion, you highlight a methodological difference between Uganda and the other two countries in defining “fishing households”. This would also seem to fit well in the methods.

Defining proximity

- Distance to markets / water bodies appears to have been included as a continuous variable, but is reported in several cases as a binary categorical variable (e.g. 118, Fig 4), with 5km or less considered close and greater than 5km distant. Could you describe how this definition / threshold was selected?
- Table 11 mentions a “5km buffer” being used to ensure anonymity. What does this mean in terms of distances, if 5km is the cut-off for being considered proximate to water?
- Also worth indicating distances to be based on straight-line distances, and considering whether this may introduce bias in any particular landscapes within the region?

Diet and food security indicators

- 295-300. Could you expand on the reasons for indicator selection? Since key contributions of fish are to protein and micronutrient intake, the reader might question why a proxy measure of energy sufficiency is used. I (and I imagine many others) am less familiar with the FCS, so highlighting its value over HDDS would be helpful. Perhaps acknowledge that although it hasn't been validated as a proxy for micronutrient intake, it does involve a weighting of nutrient-rich food groups rather than a simple count of groups (as per HDDS)?
- It seems you had access to quantitative estimates of fish intake at a household level, and these were divided by the number of household members to reach individual intake? Why not use a weighting such as the Adult Male Equivalent to account for differences in consumption by different household members, rather than assume all household members to have an equal share? e.g. <https://doi.org/10.1016/j.foodpol.2017.08.010>

Seasonality of data

- You don't appear to discuss the influence of seasonality on associations identified. Both smallholder agriculture and small-scale fishing livelihoods have a strong seasonal dimension, with both direct access to foods produced and income generated concentrated in certain months. It would be useful to indicate the timing of data collection – was this consistent across countries? Was

there a wide spread within countries? How might this affect dietary outcomes?

- Obviously the timing of seasonal food insecurity will vary between agroecological zones / livelihood strategies / according to weather events in a given year, but providing some context would increase rigour. Also advise adding a section on limitations of your study, where you could raise some of the uncertainty introduced by data availability.

Wealth index

82, 293, etc. I notice you mention the wealth index to have been established using PCA in the supplementary material, but it would be valuable to add some brief additional details of the approach taken, and relevant reference(s), in the main body of the manuscript. This would support the rigour of your wealth assessment, and also help others conducting similar analyses (I find I'm often searching for examples of how others have constructed wealth indices!) Also, what sort of access to infrastructure was included?

Statistical analysis

Throughout the results, clarity on which associations are based on t-tests and which on regression analysis could be improved – to highlight where covariates have been considered.

And some minor points:

- 24. Suggest “achieve an adequate Food Consumption Score” to provide greater clarity in abstract
- 39-40. Perhaps “policy discourse and development efforts, globally and in low-middle income countries”?
- 102. Referring to dietary assessment method would provide greater context for the patterns reported. e.g. Using household-level dietary data for a seven-day recall period, fish was identified as the dominant animal-source food...
- 106. Suggest a 1% difference doesn't warrant “marginally exceeded”, given margins of error; instead, perhaps “compared with Uganda where the number of households consuming fish and beef was approximately equal (33% and 32%, respectively).”
- 111. How do the categories of dried / fresh fish relate to species? Are these small dried fish and large fresh fish, or is there some overlap?
- 111. Nutrient composition and price are mentioned in passing, with no reference to data sources or analyses conducted. Please add to methods. I see in the supplementary material that 2010 price data was used. Why not use data relevant to the year of each country's data collection?
- 194-5. Suggest most accessible.
- 120. Poorer and wealthier based on what categories? Quintiles? Suggest clearer description, e.g. “Whilst households in the highest wealth quintile consumed twice as much fish as those in the lowest wealth quintile” ...
- 133. A food systems lens?
- 187-189. Unusual to introduce new findings in discussion. Suggest these figures would be better placed in the results.
- 249. Suggest “shape two key aspects of food environments” – since the food environment also encompasses many other dimensions beyond those considered in this analysis
- 265. typo: its
- 213 & 265. Continue alphabetical order of country names, as used elsewhere
- 215. Impact of 1% supply increase on 250,000 people based on what analysis? Add reference (or reposition reference 35 to end of sentence)

- Fig 4. Suggest adding legend for colours, rather than describing in caption.
- Fig 7. Indicates that child growth outcomes were examined, but these analyses are not presented here.
- Table 1 in Excel doc. Typo Tanzania

Reviewer #3 (Remarks to the Author):

This is a timely and important review that establishes some hard data regarding the role of small-scale fisheries in coastal community food systems and economies. It is well written, with clear and sound methods that are easy to understand. I find no noteworthy flaws in the approach or the inferences made. So, congratulations to the authorship team for assembling a really fine contribution.

I had but one recurring thought for this manuscript as I read it. While the strong dependence on fisheries is well established by the data, I feel it lacks a discussion or acknowledgment of the ecological side of the equation. What do we know about the status of the fisheries that are providing these essential services? Are they sustainably fished? Overfished by outside/industrial interests? Is there conflict over them? There are three scenarios that come to mind--first is that local governance is succeeding at sustaining the local fisheries because of tight coupling of local livelihoods to the fisheries, and hence we have a win-win / success story. It could also be that these fisheries are threatened by outside actors which threatens the food and livelihood security. The third is the marginalization/degradation hypothesis comes to mind (also, the poverty trap), in that perhaps these communities are locked into overharvesting these fisheries because of other socioeconomic circumstances (lack of alternatives, political exclusion, etc).

While it would likely be a large undertaking to do an extensive socio-ecological analysis of all the fisheries, it seems reasonable to at least offer some background on the status/sustainability, perhaps in a way that prompts & supports future, more in-depth look at the question.

Response to Reviewers

Reviewer #1 (Remarks to the Author):

The paper wishes to present relationship between fishery/poverty and food. Despite using a large database, the results are not significant from a statistical and empirical point of view. For instance the 1.2% more poverty every 10km distance from the shores is a non sens and doesn't mean something at the household level. Same with the other figures (food insecurity ... 0.3 and 0.6%). Peasants are entering the fishery when drought or other events are altering crops and most of them are seasonal fishermen (at the end of the raining season) but they do come back to their land. So, overall, due to the lack of a strong ground anchorage, the paper provides generalities that are already known and provides little for policy implication.

Authors response: on the regression results.

We fully agree with the reviewer that the key messages emerging from the regression analysis should be better clarified. At the same time, we would like to stress that the estimated coefficients associated with the distance to water bodies are statistically significant at 1% level. The statistical significance of the estimated coefficients applies to all specifications presented in the model tables: Table 1a and 1b (see standard errors and p values in Tables 1a and 1b).

Acknowledging the reviewer's comment, we are aware that the way the regression results were presented and discussed in the previous version of the manuscript does not really do justice to the importance of living in proximity to small-scale fisheries and its interconnectedness with poverty and food security. We have revised the manuscript and provided more details behind the regression results and clarified their meaning. We have also estimated the probabilities to be poor and food insecure for households living at different distances to small-scale fisheries location (i.e., household grouped by quintile of distance to water bodies), based on the coefficients of the regression analysis, to better communicate the results (Supplementary Data 1, Tables 1a & 1b). Please let us clarify three important points of the analysis which have been updated in the manuscript.

Firstly, in order to provide a better understanding behind the estimated coefficients, we have added at the margin of the regression tables (Tables 1a and 1b): 1) the mean distance to water bodies for each country; and 2) the "baseline probability" to be poor or food insecure (i.e. the probability to be poor or food insecure regardless of whether households live close or distant to water bodies).

Secondly, considering these new details added to the regression tables, the interpretation of the estimated coefficients associated with the households' distance to water bodies has now been clarified throughout the manuscript. For ease of reference, here we provide an example of the interpretation of the estimated coefficient of households' distance to water bodies on the probability to be poor (based on specification 1 in table 1a): when all countries are pooled together (first specification of the regression table), the "baseline" probability to be poor for a given household is 38.4% (0.384). This means that an increase of 1 unit (km) from the mean distance to water bodies (mean distance=33.1 km) is associated with an expected probability of being poor equal to $0.384+\beta$ (i.e., the estimated coefficient

from the probit regressions), holding all other variables constant. Although the estimated coefficients presented in the regression tables seem to be small, what is important to highlight is the dynamic behind the regression model. Indeed, the regression coefficient of the distance to water bodies is meant to capture the additive effect that is added to the baseline probability to be poor (which is already very high across the three countries), when households live at further distances from the mean distance to small-scale fisheries locations.

Thirdly, to make the regression results clearer, we have calculated household-specific probabilities to be poor and food insecure based on different distances to water bodies (household grouped by quintile of distance to water bodies). This is done by using the estimated coefficients obtained from the two probit regressions. In general, we found that, on average: the probability to be poor and food insecure is 15 and 12.6 percentage points lower for households living in proximity to small-scale fisheries (mean distance ≤ 3 km), compared to households living at the longest distance small-scale fisheries (mean distance > 79 km). The table below provides details for these probabilities for poverty. These new estimated probabilities have been added in the manuscript within the Supplementary Data Table 1a and 1b for poverty and food insecurity, as they provide a better and easier interpretation compared to the regression coefficients.

Poverty																
Quintile of distance to water bodies	All countries				Malawi				Tanzania				Uganda			
	National		Rural		National		Rural		National		Rural		National		Rural	
	Prob. to be poor	Dist. (km)	Prob. to be poor	Dist. (km)	Prob. to be poor	Dist. (km)	Prob. to be poor	Dist. (km)	Prob. to be poor	Dist. (km)	Prob. to be poor	Dist. (km)	Prob. to be poor	Dist. (km)	Prob. to be poor	Dist. (km)
1	34%	2.71	45%	3.33	45%	3.99	47%	3.11	37%	2.19	59%	3.42	19%	351%	22%	3.63
2	29%	11.82	43%	15.65	44%	20.41	54%	18.19	35%	10.47	56%	17.08	12%	1084%	17%	13.09
3	37%	26.27	45%	31.28	40%	35.76	55%	36.60	44%	25.45	59%	32.86	21%	2050%	23%	25.88
4	43%	44.65	49%	48.34	49%	50.69	54%	52.59	51%	45.50	56%	50.23	23%	3766%	25%	41.74
5	49%	79.32	57%	80.88	46%	74.33	50%	75.16	54%	85.90	67%	88.62	30%	6851%	33%	71.60
Mean	38%	33.06	48%	35.96	45%	36.95	52%	37.07	44%	33.75	59%	38.03	21%	2847%	24%	31.16
Obs.	18,623		14,283		12,444		10,174		3,344		1,978		2,822		2,125	

Text amendments:

Text has been amended throughout the manuscript, in the abstract and results to better clarify and communicate statistical results. New information has been added to the regression tables (Table 1a & 1b) and estimated probabilities have been added in the manuscript within Supplementary Data Table 1a and 1b for poverty and food insecurity (see one table below).

Authors: on seasonality in small-scale fisheries and on the paper providing generalities that are already known and provides little for policy implication.

Yes, most fishing households are fisher-farmers, where virtually all fishing households included in our sample engage in both fishing and farming activities (96.6%); which is typical in small-scale fisheries contexts. Small-scale fisheries can provide a livelihood of last resort during times of shocks, but also a viable livelihood all year round across its value-chain (Béné et al., 2016; Simmance, 2017; Simmance et al., 2021). The aim of the paper is to understand if engagement in small-scale fisheries or living in proximity to small-scale fisheries is associated with improved food security and/or reduced poverty. The survey data does not allow us to unpackage the seasonality of livelihood activities as questions on livelihoods are related to whether the household engaged in those activities over the previous 12 months. Thus, our findings demonstrate the value of small-scale fisheries livelihoods, but cannot reveal the sequential nature of livelihood development or seasonality. In addition, the surveys in our paper did collect data over a period of 12 month: for each month over the one-year period, a different portion of total household, randomly selected, was interviewed so that any indicator of food consumption will capture the intrinsic variability in food consumption which is due to the seasonality. Yet, reported food insecurity was collected using a recall period of 12 months (reported as food insecurity in number of months) whereby the surveys report the number of months to which a household does not have enough food to eat. This has demonstrated that fishing households have higher adequate food consumption profiles and experience a lower number of months of food insecurity (see Supplementary file).

Our study provides novel analyses on examining small-scale fisheries livelihoods, through the fishery module of the Living Standard Measurement Survey which to our knowledge has not been conducted before. We build on only one other known representative study in the region (Fisher et al., 2017) that investigated fish-dependent households and wealth and expand by inclusion of fish-related livelihoods across value-chains, proximity to fisheries, and more diverse poverty and food security metrics. Our findings build upon past theories and several local case studies highlighting the contribution of small-scale fisheries to poverty alleviation, but importantly does so at representative scales which is the first large study of its kind know, particularly for inland fisheries. We also provide the first representative study investigating associations between small-scale fisheries, fish consumption and food security in the region – where often representative fish consumption studies have focused on Asia. As Béné et al (2016) pointed out, many of the existing evidence is from local case studies, and more data is needed at representative national scales to truly highlight the value of small-scale fisheries and to inform policy. Our findings provide novel insights into where and for whom the distribution of benefits of fish from small-scale fisheries reach and allows policy makers and managers to be better informed on strategies to harness the potential of fish for sustainable development.

Reviewer #2 (Remarks to the Author):

Thanks for the opportunity to review your interesting manuscript. By drawing on food systems concepts and frameworks, and harnessing large nationally representative datasets, this manuscript responds to key knowledge gaps about patterns of fish intake and food security in a region with potential for

investment and innovation in small-scale fisheries to contribute to combatting the ongoing high prevalence of undernutrition. The manuscript is well-written, presents novel analyses, contextualizes these within the wider literature, and proposes their significance for research and development efforts. Please consider these specific methodological points, which I believe would increase rigor, improve clarity and acknowledge limitations:

Categorising livelihoods

- 290. Why have mixed livelihood strategies been excluded from your analysis? It is acknowledged that households commonly combine fishing and agricultural activities, which makes sense given the seasonal variation in income and labour for both. It seems unwise to exclude these households from your sample, given the value of understanding whether mixed livelihood strategies offer greater resilience to seasonal food insecurity. Why not consider these as an additional livelihood group / typology within your analyses?

Authors: on mixed livelihood strategies: This is a key comment, which deserves to be properly clarified in the manuscript. The comments have been addressed in the methods section.

We currently include 3 groups of livelihoods:

1. **Fishing households (harvesting, processing and/or trade) – households engaged in fishing only and those who engage in agriculture livelihoods as well.**
2. **Only agriculture households - non-fishing.**
3. **Neither fishing nor agriculture households.**

Households engaged in mixed livelihoods are not excluded from our analyses sample. Rather, the purpose of the study is to understand whether engagement in fish-related activities is associated with positive sustainable development, which includes households engaged in diverse livelihood portfolios. Numerous studies have examined livelihoods in relation to comparing fishing and non-fishing households (Gomma and Rana, 2007; Darling, 2014; Garaway, 2005; Fiorella et al., 2014; Simmance, 2017). We acknowledge that it would be very interesting to unpackage the diversity of portfolio of livelihood activities, particularly within the fishing household category, however we feel that this is out of the scope of this study would warrant a different approach such as cluster analyses of livelihood typologies (Dobbie, 2017). Through data exploration, small-scale fishing households in the sample engage in diverse livelihood activities where we found that about 96% of total fishing households also conduct agriculture activities in parallel or consecutively to fisheries, with less than 4 % reported to engage exclusively in small-scale fisheries. The analysis shows the following:

Type of household	%
Neither fishing nor agriculture.	24.32
Only fishing	0.1
Fishing and agriculture	2.83
Only agriculture	72.75
Total	100

Unfortunately, due to the very small number of observations capturing “pure fishing household” (0.1 % of total household), adding this extra-group to the analysis would lead to inconsistent statistics, especially when the key indicators used for the analyses (e.g., poverty, food consumption profile etc.) are to be disaggregated by typology of household. However, we have added more detail on the livelihood groups in the methods section.

Text has been amended in the methods section as follows:

“For each country survey, fishing households and agriculture households are defined as at least one member engaging in fish-related activities or agriculture for any period over the previous 12 months to which the survey was conducted. Fish-related livelihood activities were defined as fish harvesting, processing, and trading in Malawi and Tanzania, whilst in Uganda they were defined only as fishing. Through data exploration of livelihood categories, we found that 96% of all fishing households in our study combine fish-related and agricultural activities, with only 4% engaging exclusively in small-scale fisheries. Examination of diverse livelihood typologies within fishing household category (e.g. fisher-farmer, which is common in the region or exclusive fisher) was deemed out of the scope of this study due and not feasible due to the small number of observations of exclusive fishers”.

- Also worth adding a sentence or two to better define livelihood categories in the methods section. i.e. at least one household member involved in fishing / agriculture for any period of time within the past 12 months? In the discussion, you highlight a methodological difference between Uganda and the other two countries in defining “fishing households”. This would also seem to fit well in the methods.

Authors: on the definition of livelihood categories.

We have addressed this comment in the paper and defined livelihood categories (lines 298-315). For each country survey, fishing households and agriculture households are defined as at least one member engaging in fish-related activities or agriculture for any period of time over the previous 12 months to which the survey was conducted. Fish-related livelihood activities were defined as fish harvesting, processing and trading in Malawi and Tanzania, whilst in Uganda they were defined only as fishing (see Supplementary Data – Table 1).

Country	Survey & Year	Survey livelihood question	Time of year data collected
Uganda	National Panel Survey, 2010-2011	“Did anybody in the household practice A: river fishing; B: Natural freshwater pond/lake fishing; C Artificial fishpond fishing; D) swamp fishing”	Throughout the year
Tanzania	National Panel Survey, 2014-2015	"During the last 12 months, how many of these months did you, or someone in your household, engage in fishing, fish trading, or fish processing activities?" "Have you, or anyone in your household, caught any fish in the last 12 months, either for sale, preservation, or home consumption?" " Have you, or anyone in your household, engaged	Throughout the year

		in fish trading in the last 12 months?"	
Malawi	Fourth Integrated Household Survey, 2016-2017	"Did any HH member engage in fishing or fish trading"	Throughout the year

We would like to highlight that for “fishing households” in our analysis, small-scale fishery is most likely to be the main income generating activity. Indeed, a closer look at the fishery module attached to these three surveys clearly indicate that “small-scale fishing households” that are part of our study organize their fishing activities by using a complex mix of input factors (i.e., external labor input, different gear types etc.), even if they can also engage in other non-fishing activities during the off-season.

Text has been amended in the methods section as follows:

“For each country survey, fishing households and agriculture households are defined as at least one member engaging in fish-related activities or agriculture for any period over the previous 12 months to which the survey was conducted. Fish-related livelihood activities were defined as fish harvesting, processing, and trading in Malawi and Tanzania, whilst in Uganda they were defined only as fishing”.

Defining proximity

- Distance to markets / water bodies appears to have been included as a continuous variable, but is reported in several cases as a binary categorical variable (e.g., 118, Fig 4), with 5km or less considered close and greater than 5km distant. Could you describe how this definition / threshold was selected?
- Table 11 mentions a “5km buffer” being used to ensure anonymity. What does this mean in terms of distances, if 5km is the cut-off for being considered proximate to water?
- Also worth indicating distances to be based on straight-line distances, and considering whether this may introduce bias in any particular landscapes within the region?

Authors: on the calculation of distance to water bodies.

Thank you for this comment. We have clarified in the manuscript (methods section). For our descriptive statistics the cut-off threshold of 5 Km was used to compare poverty and food security indicators among households proximate and distant (≤ 5 km was considered close and >5 km was considered far) from fished water bodies, and between fishing and non-fishing households (see full details in Supplementary Data 1). The cut-off threshold reflects the need to provide a comparison of the key indicators used throughout the manuscript between groups and check whether potential differences between those two groups of households are statistically significant. The significance of differences between groups was finally tested using the T-test.

The selection of the 5 km cut-off threshold to classify households as distant or proximate to water bodies was both guided by other scientific papers (Temsah et al., 2018; O’Meara et al., 2021; Lo et al., 2019; Alva et al., 2018) and based on the distribution of households by quintile of distance to water bodies. As reported in the table below, in all countries, the bottom 20 % of households with the shortest distance to water bodies live at less than 5 km, on average. In order to compare distant and proximate households in a selection of three countries, we therefore have selected a cut-off distant, based on the

first quintile, which would enable cross-country comparison. Thus, household proximate to water bodies are those having a distant to water bodies which is no longer than 5 Km.

Quintile of distance	Km	Malawi	Tanzania	Uganda
1	mean	2.37	2.57	3.25
	max	6.30	6.33	6.32
2	mean	12.23	11.74	11.87
	max	17.37	17.47	17.25
3	mean	26.35	26.41	25.87
	max	34.71	34.57	34.58
4	mean	44.20	44.84	44.74
	max	54.50	54.48	54.49
5	mean	70.25	84.09	77.87
	max	160.71	154.78	172.91

For the regression analyses, we preferred to include the distance to water bodies as a continuous variable. This was done since we wanted to measure the marginal increase in the probability to be poor or food insecure given a one unit increase in the distance to water bodies from its mean. We believe that, when the distance to water bodies is included in probit regressions as a continuous variable, we can get a better understanding of dynamics related to households that to live more distant from small-scale fisheries locations. Indeed, by including a continuous variable in the probit regression, we could measure the additive effect to the baseline probability to be poor (which is already very high across the three countries), when households live at longer distances to small-scale fisheries locations. This latter would haven't been possible if the regression model had a "crude" distinction between "proximate" and "distant" households.

Authors: on the "5km buffer" being used to ensure anonymity.

This comment is very relevant, and it has been clarified in the manuscript (methods section).

To ensure anonymity of sampled households, GPS coordinates have been slightly modified before being disseminated. The modification strategy relies on random offset of cluster center-point within a specified range determined by an urban/rural classification. For urban areas a range of 0-2 km is used. In rural areas, where communities are more dispersed and risk of disclosure may be higher, a range of 0-5 km offset is used. This implies that, in the worst-case scenario, a household that lives 5 km to a given water bodies, could be 10 km distant to the same water bodies. Unfortunately, we have no control over this as it was already applied to the dataset available for analyses. However, considering that, the modification strategy relies on a random offset which is applied to both distant and proximate household, the distribution between households close and distant to water bodies has remained unchanged, with not expected biases in the statistics presented.

Authors: on potential biases introduced by the calculation of the straight-line distance.

This is indeed a potential data limitation. We have addressed it in the manuscript (methods section).

“We acknowledge two limitations behind the calculation of the straight-line distance to water bodies. First, using the straight-line distance to water bodies may introduce biases in the statistical analyses presented, especially for households located in any particular landscapes within the country. Although, the walking or travel time distance over a road network would provide a better alternative., however there is lack of data on road networks. Despite the straight-line distance to water bodies encompasses some limitations, we still believe that this method of calculation provides a good proxy to categorize household in relation to their distance to water bodies, and the results from the analyses should not deviate substantially from other method of calculation. For example a study found that the straight-line distance tends to be highly correlated ($R>0.91$) with both walking and travel time distance.

Second, an additional bias in the presented analyses may be introduced due to the modification strategy of the households GPS coordinates. This strategy was implemented before dissemination of household level data to avoid the risk of disclosure of sampled households. In its essence, the modification strategy relies on random offset of cluster center-point within a specified range. For urban areas a range of 0-2 km is used. In rural areas, where risk of disclosure may be higher due to more dispersed communities, a range of 0-5 km offset was used. While we had no control over this modification strategy, we believe that the modification of the GPS coordinates does not affect the way households are classified in relation to their distance to fishing grounds: considering that the modification strategy was applied to both distant and proximate households, we expect that the distribution between households close and distant to water bodies has remained unchanged and, hence, the presented statistics are still valid for the analysis”.

Diet and food security indicators

- 295-300. Could you expand on the reasons for indicator selection? Since key contributions of fish are to protein and micronutrient intake, the reader might question why a proxy measure of energy sufficiency is used. I (and I imagine many others) am less familiar with the FCS, so highlighting its value over HDDS would be helpful. Perhaps acknowledge that although it hasn't been validated as a proxy for micronutrient intake, it does involve a weighting of nutrient-rich food groups rather than a simple count of groups (as per HDDS)?

Authors: on the reasons for indicator selection.

The following text has been added/amended into the methods section:

From line 390: “The LSMS-ISA surveys collects food consumption data over a 7-day recall period throughout the year to capture seasonality. We used the FCS index as a food security indicator as it is akin to the data collected via the LSMS-ISA surveys, and that there was a need for comparison across select countries. The FCS index measures the frequency (number of days) and diversity of food groups consumed over a 7-day recall period, with weights given to groups based on nutritional value. The FCS

index is validated as a proxy for energy sufficiency (quantity of food) and food access, and is associated with other household-level diet diversity measures (e.g. household dietary diversity score (HDDS))^{20,49}. The difference between FCS and other indicators such as HDDS is the recall period (7-days versus 24 hour), diversity of groups, weights assigned based on nutrition, and use of frequency together with diversity of groups consumed. The FCS with a longer recall period can show more habitual consumption but can also have limitations with people's recall reliability. Although it has not been validated yet as an indicator for micronutrient intake, it does provide weights to nutrient-rich food groups and accounts for frequency of consumption, which other indicators do not".

- It seems you had access to quantitative estimates of fish intake at a household level, and these were divided by the number of household members to reach individual intake? Why not use a weighting such as the Adult Male Equivalent to account for differences in consumption by different household members, rather than assume all household members to have an equal share? e.g. <https://doi.org/10.1016/j.foodpol.2017.08.010>

Authors: on the quantitative estimates of fish intake at a household level.

We fully agree with the reviewer that per-capita fish and other animal source food intake should be estimated using an adult equivalent scale. As the main purpose of our paper is to examine household-level indicators, and not the fish intake at level of individual within the household, we have decided to remove the text below. On reflection of the results and findings, individual level estimates do not add to the discussion above what is already revealed and highlighted from the household level data, and individual level data has its limitations as intra-household distribution of consumption is not examined.

Lines 169+ removed: "At the national level, average quantities of fish consumed met the EAT Lancet universal recommendation of fish consumption for a sustainable and healthy diet (28 g of fish per day, range of 0-100 g) (30 g/capita/day in Malawi, 29 g in Tanzania and 36 g in Uganda)²¹ (Supplementary Data 1, Table 9)"

Lines 248+ edited to read as follows: "Rural and poor households distant from fishing grounds consumed some of the lowest quantities of fish. However, dried fish were found to be accessible to these remote populations and were often the main accessible animal-source food".

Seasonality of data

You don't appear to discuss the influence of seasonality on associations identified. Both smallholder agriculture and small-scale fishing livelihoods have a strong seasonal dimension, with both direct access to foods produced and income generated concentrated in certain months. It would be useful to indicate the timing of data collection – was this consistent across countries? Was there a wide spread within countries? How might this affect dietary outcomes?

Authors:

The LSMS datasets collect data throughout the previous year (12 months), so it does account for seasonality, and provides an average value of the indicator of interest. For example, in Malawi, sampled

households (in total 12,447) were interviewed from April 2016 to April 2017, see table below, and each month over the data collection period a different portion of households was interviewed.

Depending on the recall period of the indicator of interest (which is 7 days for food/fish consumption) and given the distribution of the sample by months of data collection, the constructed indicators, for example fish consumption, is assumed to be a robust estimate of the average quantity of fish consumed in the previous 12 months/survey collection period. The estimated average should therefore account both the lower quantity due to potential off-season and the higher quantity due to potential high seasons. Text added in the methods section as follows: “The LSMS-ISA surveys collects food consumption data over a 7-day recall period. To capture seasonal variation in the food consumption indicators, sampled households were interviewed over a 12-month period: for each month of the year, a different portion of sampled households was interviewed so that the derived indicators reflect the intrinsic variability in food consumption, which may be due to seasonality”.

interviewDate	Sample Malawi 20212
16-apr	354
16-mag	977
16-giu	1186
16-lug	575
16-ago	78
16-set	1175
16-ott	1015
16-nov	228
16-dic	554
17-gen	761
17-feb	1672
17-mar	1906
17-apr	1961
17-mag	5
Total	12447

Wealth index

82, 293, etc. I notice you mention the wealth index to have been established using PCA in the supplementary material, but it would be valuable to add some brief additional details of the approach taken, and relevant reference(s), in the main body of the manuscript. This would support the rigour of your wealth assessment, and also help others conducting similar analyses (I find I’m often searching for examples of how others have constructed wealth indices!) Also, what sort of access to infrastructure was included?

Authors: We have clarified how the assets wealth was constructed (methods section). Please see below the type of assets owned by the households, including the way they were combined to obtain the wealth index.

We use Principal Component Analysis (PCA), as a statistical technique for data reduction to calculate wealth index score to rank household based on their wealth status. Considering that the wealth status of a given household is driven by its typology and number of assets, these are combined in a wealth index using PCA: total number of assets owned by the household (durable goods - radio, bicycle, TV; utilities and infrastructure – access to protected water source and electricity). The technique allows constructing a series of uncorrelated linear combinations of the assets variables that contain most of the variance (principal components). This reduces the multi-dimensionality of the assets variables, and it allows the data to identify the linear combinations of the assets components that explain the greatest share of the variation in wealth. An index value of the wealth status of each household in our sample is finally derived using the estimated coefficients of the first principal component as weights. Hence, this index of wealth— i.e., the wealth status of households- is based on the type and quantity of assets owned by the household. As the final wealth index is standardized, it provides a ranking of household reflecting their ownership of assets.

Statistical analysis

Throughout the results, clarity on which associations are based on t-tests and which on regression analysis could be improved – to highlight where covariates have been considered.

Authors: throughout the manuscript, we have improved clarity on which associations are based on descriptive statistics and which are based on regression analyses.

For descriptive statistics we have specified the t-test (T-test, $p < 0.05$).

For regressions –we have specified in the narrative the β coefficient and its related p-values ($\beta = X.XXX$, $p < 0.00X$). Yet, when statistics based on regression analysis, we have introduced the statement “ceteris paribus/holding all other variable constant”, to better clarify that those estimates are based on conditional covariates. In addition to that, we have added a specific section to better elaborate on the econometric estimates (i.e., “Econometric model”). In that section, we have clarified how the model to estimate the marginal probability to be poor (and food insecure) was constructed, as well as the full range of explanatory variables that were inputted in the model to arrive to the estimated marginal effects. The full list of explanatory variables includes 1) distance to water bodies (in Km) and distance to food market (in Km). 2) An interaction variable which measured the household’s distance to water bodies but restricted to only those households who were unable to reach the food market. 3) a categorical variable that classified households according to their main livelihood activity, namely a) neither fishing, nor agriculture households (i.e., the reference baseline household category), b) fishing households and c) agriculture households. That variable was interacted with a dummy variables measuring households living in proximity to water bodies to better measure for which typology of household the proximity to fishing grounds is most beneficial. Finally, we also listed all control variables, e.g., age, sex and level of education of the head of the household, household size etc.

And some minor points:

- 24. Suggest “achieve an adequate Food Consumption Score” to provide greater clarity in abstract
- 39-40. Perhaps “policy discourse and development efforts, globally and in low-middle income countries”?
- 102. Referring to dietary assessment method would provide greater context for the patterns reported. e.g., Using household-level dietary data for a seven-day recall period, fish was identified as the dominant animal-source food...
- 106. Suggest a 1% difference doesn’t warrant “marginally exceeded”, given margins of error; instead, perhaps “compared with Uganda where the number of households consuming fish and beef was approximately equal (33% and 32%, respectively).”

All recommendations above taken on board in manuscript and text updated.

- 111. How do the categories of dried / fresh fish relate to species? Are these small dried fish and large fresh fish, or is there some overlap?

Text added as follows: Although no information is provided on fish species in the surveys, small fish species are known to be dried in the region due to their size which allows efficient drying in time, whilst larger fish species are provided fresh, or either smoked or salted.

- 111. Nutrient composition and price are mentioned in passing, with no reference to data sources or analyses conducted. Please add to methods. I see in the supplementary material that 2010 price data was used. Why not use data relevant to the year of each country’s data collection?

Authors: We have added additional information on the nutritional value and price data within the methods section. Please also see a note below to explain the price conversion process.

We agree that the method for prices construction must be clarified in the manuscript, and we thank the referee. However, since price data were estimated through surveys conducted in different years and across different countries, we believe it is important to control for both spatial and time difference between countries, in order to provide spatial- and time-deflated cost of food items. To this aim, estimated prices per Kg which were obtained using survey data in local currency unit were 1) converted in international USD, using the PPP conversion factor and 2) deflated to control for potential inflation patterns within the three countries (since surveys refer to different years).

In more detail, the conversion of prices from local currency unit to international USD was done using the Purchase power parity conversion factor corresponding to the year of the survey. The PPP is a spatial price deflator and currency converter that is used to control for price level differences between countries. A useful example behind the need to convert prices in PPP is provided by the World Bank: “suppose that there is a basket of goods and services that costs 50 United States dollars (USD). 50 USD would be equivalent to 363 South African Rand (ZAR) when using a market exchange rate of 7.26. However, due to South Africa’s lower price level in relation to the United States, the cost of a similar basket is actually 239 ZAR. Therefore, 50 USD would buy a larger basket of goods and services in South

Africa than it would in the United States; the PPP of South Africa to the United States would be 239 ZAR/50 USD, which is equal to 4.77”.

Further to the conversion into international USD dollar, we also control for potential inflation patterns within each country analyzed. This was done to provide a measure of the real cost of 1KG of fish (and other animal source food) in countries where prices data were collected in different years. Thus, the estimated nominal prices were adjusted using the consumer price index (CPI, base year=1 in 2010), i.e., a measure of the average change of prices over time. Since the three analyzed surveys were conducted in different years, nominal food prices in Malawi, even if expressed in PPP, are not directly comparable with food prices in Tanzania or Uganda, due to different inflation patterns. For example, if we assume that fish prices in Malawi 2016 were equal to fish prices in Tanzania in 2014, say 3 USD international dollar in each country (for example), but inflation has grown at faster pace in Malawi compared to Tanzania, then the 1Kg of fish would be more expensive in Malawi 2016 compared to Tanzania 2014.

The CPI deflator of the World Bank (figure below) shows that the general level of prices in Malawi has increased by 3.05 times from 2010 to 2016 (2016 year of the survey); in Tanzania the level of prices over the period 2010-2014 (2014 year of the survey) has increased by only 1.49 times.

In order to compare the real cost of 1kg of fish (and other animal source food) across countries, we needed to control for the inflation pattern in the two countries, so to derive the real price to buy the same quantity of fish in both Malawi and Tanzania. Considering the inflation pattern observed in the two countries, real price per Kg will be respectively 6.15 USD in Malawi (i.e., 2016 price=3* CPI=2.4) and 4.46 USD (i.e., 2014 price=3* CPI=1.49), which is due to higher inflation pattern in Malawi. The chart below shows the dynamic in prices in Malawi, Tanzania and Uganda (source: World Bank WDI)

- 120. Poorer and wealthier based on what categories? Quintiles? Suggest clearer description, e.g., “Whilst households in the highest wealth quintile consumed twice as much fish as those in the lowest wealth quintile”...
- 133. A food systems lens?
- 187-189. Unusual to introduce new findings in discussion. Suggest these figures would be better placed in the results.
- 249. Suggest “shape two key aspects of food environments” – since the food environment also encompasses many other dimensions beyond those considered in this analysis
- 265. typo: its
- 213 & 265. Continue alphabetical order of country names, as used elsewhere
- 215. Impact of 1% supply increase on 250,000 people based on what analysis? Add reference (or reposition reference 35 to end of sentence) – added: (Supplementary Data 1, Table 9)
- Fig 4. Suggest adding legend for colours, rather than describing in caption.
- Fig 7. Indicates that child growth outcomes were examined, but these analyses are not presented here.
- Table 1 in Excel doc. Typo Tanzania

All recommendations above taken on board in manuscript and text updated.

Reviewer #3 (Remarks to the Author):

This is a timely and important review that establishes some hard data regarding the role of small-scale fisheries in coastal community food systems and economies. It is well written, with clear and sound methods that are easy to understand. I find no noteworthy flaws in the approach or the inferences made. So, congratulations to the authorship team for assembling a really fine contribution.

I had but one recurring thought for this manuscript as I read it. While the strong dependence on fisheries is well established by the data, I feel it lacks a discussion or acknowledgment of the ecological side of the equation. What do we know about the status of the fisheries that are providing these essential services? Are they sustainably fished? Overfished by outside/industrial interests? Is there conflict over them? There are three scenarios that come to mind--first is that local governance is succeeding at sustaining the local fisheries because of tight coupling of local livelihoods to the fisheries, and hence we have a win-win / success story. It could also be that these fisheries are threatened by outside actors which threatens the food and livelihood security. The third is the marginalization/degradation hypothesis comes to mind (also, the poverty trap), in that perhaps these communities are locked into overharvesting these fisheries because of other socioeconomic circumstances (lack of alternatives, political exclusion, etc).

Authors: please see additional text added to methods:

Line 318 – “Small-scale fisheries, particularly inland fisheries, in this region are known to be highly productive with a linear increasing trend in catches over the last three decades (Funge-smith, 2018; Kolding et al., 2019). On average 70% of the total catches consist of small pelagic species, which are largely driven by climate, and are highly productive, resilient, and under-exploited (Kolding et al., 2019).

However, challenges do exist in fisheries governance and signs of over-exploitation of some few fish stocks (Jul-larsen et al, 2003), as well as high post-harvest fish waste and loss across value-chains which undermine the potential benefits from the sector (Simmance et al., 2021)".

The discussion already includes text in relation to harnessing the lightly exploited stocks and accounting for challenges in the sector.

12th May 22

Dear Dr Simmance,

Your manuscript titled "Proximity to small-scale inland and coastal fisheries is associated with improved income and food security" has now been seen by our reviewers, whose comments appear below. In light of their advice I am delighted to say that we are happy, in principle, to publish a suitably revised version in Communications Earth & Environment under the open access CC BY license (Creative Commons Attribution v4.0 International License).

We therefore invite you to revise your paper one last time to address the remaining concerns of our reviewers. In particular, please carefully consider the comments of Reviewer #1 and modify your wording to ensure it is clear and unambiguous. At the same time we ask that you edit your manuscript to comply with our format requirements and to maximise the accessibility and therefore the impact of your work.

EDITORIAL REQUESTS:

Please review our specific editorial comments and requests regarding your manuscript in the attached "Editorial Requests Table". Please outline your response to each request in the right hand column. Please upload the completed table with your manuscript files.

SUBMISSION INFORMATION:

OPEN ACCESS:

Communications Earth & Environment is a fully open access journal. Articles are made freely accessible on publication under a [CC BY license](http://creativecommons.org/licenses/by/4.0) (Creative Commons Attribution 4.0 International License). This license allows maximum dissemination and re-use of open access materials and is preferred by many research funding bodies.

For further information about article processing charges, open access funding, and advice and support from Nature Research, please visit <https://www.nature.com/commsenv/article-processing-charges>

At acceptance, you will be provided with instructions for completing this CC BY license on behalf of all authors. This grants us the necessary permissions to publish your paper. Additionally, you will be asked to declare that all required third party permissions have been obtained, and to provide billing

information in order to pay the article-processing charge (APC).

[link redacted]

Best regards,

Joe Aslin
Senior Editor,
Communications Earth & Environment
<https://www.nature.com/commsenv/>
Twitter: @CommsEarth

REVIEWERS' COMMENTS:

Reviewer #1 (Remarks to the Author):

Despite changes in the text the main issue remains: no statistical meaningfulness of results.

"We found (Table 1a) that when the distance from fishing grounds increases by 1km from its mean (33.1Km), the prevalence of poverty is expected to increase by 0.14 percentage points, given a "baseline" probability of being poor equal to 38.4% in the three countries (baseline probability for poverty 0.3839; $\beta = 0.00136$, $p < 0.01$. Table 1a), holding all other variables constant"

If we read this correctly the more I'm away from the coast the more i'm poor. Fundamentally incorect, otherwise all people will live on the coast....

Reviewer #2 (Remarks to the Author):

Dear authors, thanks for your considerable time and effort in responding to comments and incorporating suggestions. I believe these have improved clarity, transparency and rigor. I've just attached one further suggestion for phrasing of fishing/agriculture households (for consideration) and noticed one typo. Congratulations on this comprehensive and valuable piece of work!

Reviewer #3 (Remarks to the Author):

Thank you for this revision. My concerns were clearly addressed and I am impressed by the nuanced attention to the other two reviews.

On mixed livelihood strategies:

Thanks for this clarification – this makes much more sense! Please consider the suggested edits below which might further clarify that ‘fishing households’ include fishing and fishing + agriculture; and ‘agriculture households’ are agriculture only.

“For each country survey, ~~fishing households were categorised according to their engagement in fishing and/or agriculture activities in the prior 12 months. Households in which one or more member and agriculture households are defined as at least one member engaged in fish-related activities were defined as ‘fishing households’ or agriculture for any period over the previous 12 months to which the survey was conducted.~~ Fish-related livelihood activities were defined as fish harvesting, processing, and trading in Malawi and Tanzania, whilst in Uganda they were defined only as fishing. Households with one or more member engaged in agriculture, but not in fish-related activities, were defined as ‘agriculture households’. Through data exploration of livelihood categories, we found that 96% of all fishing households in our study combine fish-related and agricultural activities, with only 4% engaging exclusively in small-scale fisheries. Examination of diverse livelihood typologies within fishing household category (e.g. fisher-farmer, which is common in the region or exclusive fisher) was deemed out of the scope of this study due and not feasible due to the small number of observations of exclusive fishers”.

Defining proximity:

It’s helpful to understand your approach beyond following other studies. I’m familiar with this threshold for proximity used in studies you cite, but have wondered about its definition. Distances are obviously relative to transport access – but using the first (=nearest) quintile is an useful way to define proximity.

On 5km buffer and straight line differences:

Thanks – it’s useful to add acknowledgement of both these limitations. (Just a small typo / sentence structure issue in the sentence below, i.e. use of although and however).

~~Although, t~~The walking or travel time distance over a road network would provide a better alternative, however there is lack of data on road networks.

Response to Reviewer's comments and Editorial request table:

Editorial request table

Edits have been made to the format and files in line with the checklist and editorial request table.

Reviewer #1 (Remarks to the Author):

Despite changes in the text the main issue remains: no statistical meaningfulness of results.

"We found (Table 1a) that when the distance from fishing grounds increases by 1km from its mean (33.1Km), the prevalence of poverty is expected to increase by 0.14 percentage points, given a "baseline" probability of being poor equal to 38.4% in the three countries (baseline probability for poverty 0.3839; $\beta = 0.00136$, $p < 0.01$. Table 1a), holding all other variables constant"

If we read this correctly the more I'm away from the coast the more i'm poor. Fundamentally incorrect, otherwise all people will live on the coast....

The relationship we found between distance to waterbodies/coastlines is indeed correct based on our statistical analysis, and it is important to report the exact findings. We do not expect however, that a relationship of this magnitude (i.e., small) would drive much behaviour change, such as a move to be closer to a waterbody or coastline. We also discuss this finding in greater detail in the discussion section.

We have modified our wording in the results and discussion section to ensure it is clear and unambiguous as follows in red:

➤ Results - lines 69-77:

"We found (Table 1a) that when the distance from **fished water bodies** increases by 1km from its mean (33.1Km), the prevalence of **income** poverty is expected to increase by 0.14 percentage points, given a "baseline" probability of being poor equal to 38.4% in the three countries (baseline probability for poverty 0.3839; $\beta = 0.00136$, $p < 0.01$. Table 1a), holding all other variables constant. **While this 'per kilometre' difference is small**, this translates into an estimated probability of being income poor for households proximate to water bodies that is 15.2 percentage points lower than households living distant from fishing grounds (average distance of 79.3 Km) (Supplementary Data 1, Table 1a), controlling for covariates. **While this result presents a correlation rather than indicating causality, it is consistent with other results from the analysis.**"

➤ Discussion – lines 168-174:

"Results provide new insights into how proximity to fishing grounds and engagement in fishing activities influence rural economies and contribute to SDG 1 – no poverty. Households living close to fishing grounds were able to spend more money to meet their basic needs. Fishing grounds were largely located in rural areas, with the flow of economic benefits from small-scale fisheries amplified in these contexts. **The association between fishing grounds and prevalence of**

poverty was however small, and our study did not investigate the many other socio-economic drivers of poverty and geo-location determinants (e.g., agency, mobility, land access, socio-culture, infrastructure) that influence wealth generation near fishing grounds and how those living far away exit poverty”.

Reviewer #2 (Remarks to the Author):

Dear authors, thanks for your considerable time and effort in responding to comments and incorporating suggestions. I believe these have improved clarity, transparency and rigor. I've just attached one further suggestion for phrasing of fishing/agriculture households (for consideration) and noticed one typo. Congratulations on this comprehensive and valuable piece of work!

Thank you very much for your final feedback. We have now addressed the typo and incorporated your suggested wording change to the manuscript – please see tracked changes in revised version.

Reviewer #3 (Remarks to the Author):

Thank you for this revision. My concerns were clearly addressed and I am impressed by the nuanced attention to the other two reviews.

Thank you very much for your final feedback.